# Multiplexed single-cell transcriptional response profiling to define cancer vulnerabilities and therapeutic mechanism of action

James M. McFarland [1,11], Brenton R. Paolella [1,11], Allison Warren[1], Kathryn Geiger-Schuller [1,2], Tsukasa Shibue[1], Michael Rothberg[1], Olena Kuksenko[1,2], William N. Colgan [1], Andrew Jones[1], Emily Chambers[1], Danielle Dionne[1,2], Samantha Bender[1], Brian M. Wolpin[3,4,5], Mahmoud Ghandi [1], Itay Tirosh[2,6], Orit Rozenblatt-Rosen[1,2], Jennifer A. Roth[1], Todd R. Golub [1,3,7,8], Aviv Regev [1,2,8,9,10], Andrew J. Aguirre [1,3,4,5,12✉], Francisca Vazquez [1,12✉] & Aviad Tsherniak [1,12✉]

Assays to study cancer cell responses to pharmacologic or genetic perturbations are typically restricted to using simple phenotypic readouts such as proliferation rate. Information-rich assays, such as gene-expression profiling, have generally not permitted efficient profiling of a given perturbation across multiple cellular contexts. Here, we develop MIX-Seq, a method for multiplexed transcriptional profiling of post-perturbation responses across a mixture of samples with single-cell resolution, using SNP-based computational demultiplexing of single-cell RNA-sequencing data. We show that MIX-Seq can be used to profile responses to chemical or genetic perturbations across pools of 100 or more cancer cell lines. We combine it with Cell Hashing to further multiplex additional experimental conditions, such as post-treatment time points or drug doses. Analyzing the high-content readout of scRNA-seq reveals both shared and context-specific transcriptional response components that can identify drug mechanism of action and enable prediction of long-term cell viability from short-term transcriptional responses to treatment.

[1] Broad Institute of MIT and Harvard, Cambridge 021242 MA, USA. [2] Klarman Cell Observatory, Broad Institute of MIT and Harvard, Cambridge 021242 MA, USA. [3] Harvard Medical School, Boston 02115 MA, USA. [4] Brigham and Women's Hospital, Boston 02115 MA, USA. [5] Department of Medical Oncology, Dana Farber Cancer Institute, Boston 02115 MA, USA. [6] Department of Molecular Cell Biology, Weizmann Institute of Science, Rehovot, Israel. [7] Department of Pediatric Oncology, Dana Farber Cancer Institute, Boston 02115 MA, USA. [8] Howard Hughes Medical Institute, Chevy Chase 20815 MD, USA. [9] Koch Institute of Integrative Cancer Research, Cambridge 021242 MA, USA. [10] Department of Biology, MIT, Cambridge 021242 MA, USA. [11] These authors contributed equally: James M. McFarland, Brenton R. Paolella. [12] These authors jointly supervised this work: Andrew J. Aguirre, Francisca Vazquez, Aviad Tsherniak. ✉email: andrew_aguirre@dfci.harvard.edu; vazquez@broadinstitute.org; aviad@broadinstitute.org

arge-scale screens of chemical and genetic vulnerabilities across hundreds of cancer cell lines are important for identifying new therapeutic targets and are providing key insights into cancer biology and gene function[1–7]. However, the ability of these approaches to reveal the cellular mechanisms and pathways underlying such cancer vulnerabilities is typically limited by their reliance on a single readout of cell viability to assess the effects of each perturbation.

In contrast, high-content readouts may provide opportunities to capture a more detailed picture of the cellular effects of a perturbation that underlie an observed fitness effect or arise independently of any observable fitness effects[8–11]. In particular, expression profiles are a robust and informative phenotypic measure of cellular responses to perturbations, with applications such as identifying drug mechanism of action (MoA), gene function, and gene regulatory networks[8–13]. High-throughput gene expression profiling in a limited number of contexts[10,14,15] has been used to produce large datasets of perturbation signatures—most notably the Connectivity Map (CMAP, ref. [10])—enabling systematic analysis of the space of transcriptional responses across perturbations.

Until recently, however, such assays required each perturbation or cell type to be profiled separately, limiting their cost-effectiveness and broader adoption. In particular, previous efforts have largely focused on studying responses in a small number of cell line contexts. However, perturbation responses are often context-specific, reflecting their interaction with the cell's underlying genomic or functional features. For example, targeted drugs may elicit responses only in cell lines harboring particular oncogenic mutations or expressing certain genes, making observed results specific to the particular cell line models chosen[4,6,7,16,17]. More generally, the inability to efficiently measure transcriptional responses across diverse cell contexts has limited our understanding of how perturbation effects differ across genomic and molecular cell states, which could be critical for predicting the therapeutic response of patient tumors.

The recent advent of single-cell genomics[18,19] and development of methods for profiling cell viability in pooled cell cultures[20] could together help address these challenges. In parallel, new assays, such as Perturb-Seq[8,9], have combined pooled perturbation screens with a single-cell RNA-Seq (scRNA-seq) readout and could thus provide the necessary scale and resolution to assay many cells within a mixed culture. A very recent study[21] used massively parallel scRNA-seq with combinatorial barcoding to profile the responses of a few individual cell lines to diverse drugs and doses. However, existing studies have not yet tackled profiling the responses of many diverse cell lines to a given perturbation in order to assess context-specific effects.

Here we facilitate the study of posttreatment gene expression signatures across multiple cell lines in parallel by developing MIX-Seq: Multiplexed Interrogation of gene eXpression through single-cell RNA Sequencing. MIX-Seq combines (i) the ability to pool hundreds of cancer cell lines and co-treat them with one or more perturbations and (ii) the power of scRNA-seq to simultaneously profile the cells' responses and resolve the identity of each cell based on single-nucleotide polymorphism (SNP) profiles. We show that MIX-Seq enables efficient study of transcriptional signatures after pharmacologic or genetic perturbation, evaluation of temporal evolution of post-perturbation transcriptional response, investigation of the MoA of novel small-molecule compounds, and development of novel therapeutic response prediction methods in cancer cell models.

## Results
**Multiplexed transcriptional profiling using scRNA-seq**. MIX-Seq uses scRNA-seq to measure the transcriptional effects of a perturbation across diverse cancer cell lines cultured and perturbed in one or more pools (Fig. 1a). Specifically, we co-culture cancer cell lines in pools and treat them with a small-molecule compound (or genetic perturbation)[20]. To ascertain transcriptional response signatures, cell-specific transcriptomes are measured using scRNA-seq after a defined time interval following perturbation. To assign each profiled cell to its respective cell line, we created an optimized computational demultiplexing method that classifies cells by their genetic fingerprints, similar to recently developed methods such as Demuxlet[22].

Specifically, for each single cell we estimate the reference cell line whose genotype across a panel of commonly occurring SNPs would most likely explain the observed pattern of mRNA SNP reads (Fig. 1b). As previously demonstrated, this also allows for identification of multiplets of co-encapsulated cells[22], where two or more cells from different cell lines are unintentionally tagged with the same cell barcode during droplet-based single-cell library preparation. Our pipeline utilizes a fast approximation strategy to identify such "doublets" that efficiently scales to pools of hundreds of cell lines ("Methods"). It also provides quality metrics that can be used to identify and remove low-quality cells (Supplementary Fig. 1), such as empty droplets[19,23].

We confirmed the classification accuracy of our SNP-based demultiplexing in two ways. First, we classified cell identities based on either their gene expression or SNP profiles ("Methods"), finding that these independent classifications were in excellent (>99%) agreement (Supplementary Fig. 2). While either feature could thus be used to accurately classify cell identities, we focus on SNP-based classification here, as it is inherently robust to perturbations that could dramatically alter the cells' expression profiles and could be applied to pools of primary cells of the same type from different individuals (e.g., induced pluripotent stem cells). Second, we allowed the SNP classification model to select from a much larger panel of 494 reference cell lines (Supplementary Data 1) and assessed the frequency with which it identified cell lines that were not in the experimental pools. The model never picked an out-of-pool cell line (0/84,869 cells passing quality control (QC)). Notably, though we tested MIX-Seq with experimental pools of up to 99 cell lines, these analyses show that SNP profiles can be used to distinguish among much larger (>500) cell line pools. Furthermore, downsampling analysis showed that SNP-based cell classifications can be applied robustly to cells with as few as 50–100 detected SNP sites (Supplementary Fig. 3).

**MIX-Seq identifies selective perturbation responses and MoA**. Next, we evaluated whether MIX-Seq could distinguish biologically meaningful changes in gene expression in the context of drug treatment. We treated pools of well-characterized cancer cell lines with 13 drugs, followed by scRNA-seq at 6 and/or 24 h after treatment (Supplementary Data 2). These included eight targeted cancer therapies with known mechanisms, four compounds that broadly kill most cell lines, and one tool compound (BRD-3379) with unknown MoA that was found to induce strong selective killing in a high-throughput screen. In all cases, we compared our scRNA-seq-based phenotyping to long-term viability responses measured for these drugs and cell lines from the genomics of drug sensitivity in cancer (GDSC) dataset[4,7], as well as data generated using the PRISM assay[17,20] ("Methods").

As a benchmark, we first consider nutlin, a selective *MDM2* inhibitor, which we applied to a pool of 24 cell lines. *MDM2* is a negative regulator of the tumor-suppressor gene *TP53*, and nutlin is known to elicit rapid apoptosis and cell cycle arrest exclusively in cell lines that have wild-type (WT) *TP53*[24]. Jointly embedding the expression profiles of 7317 single cells treated with either

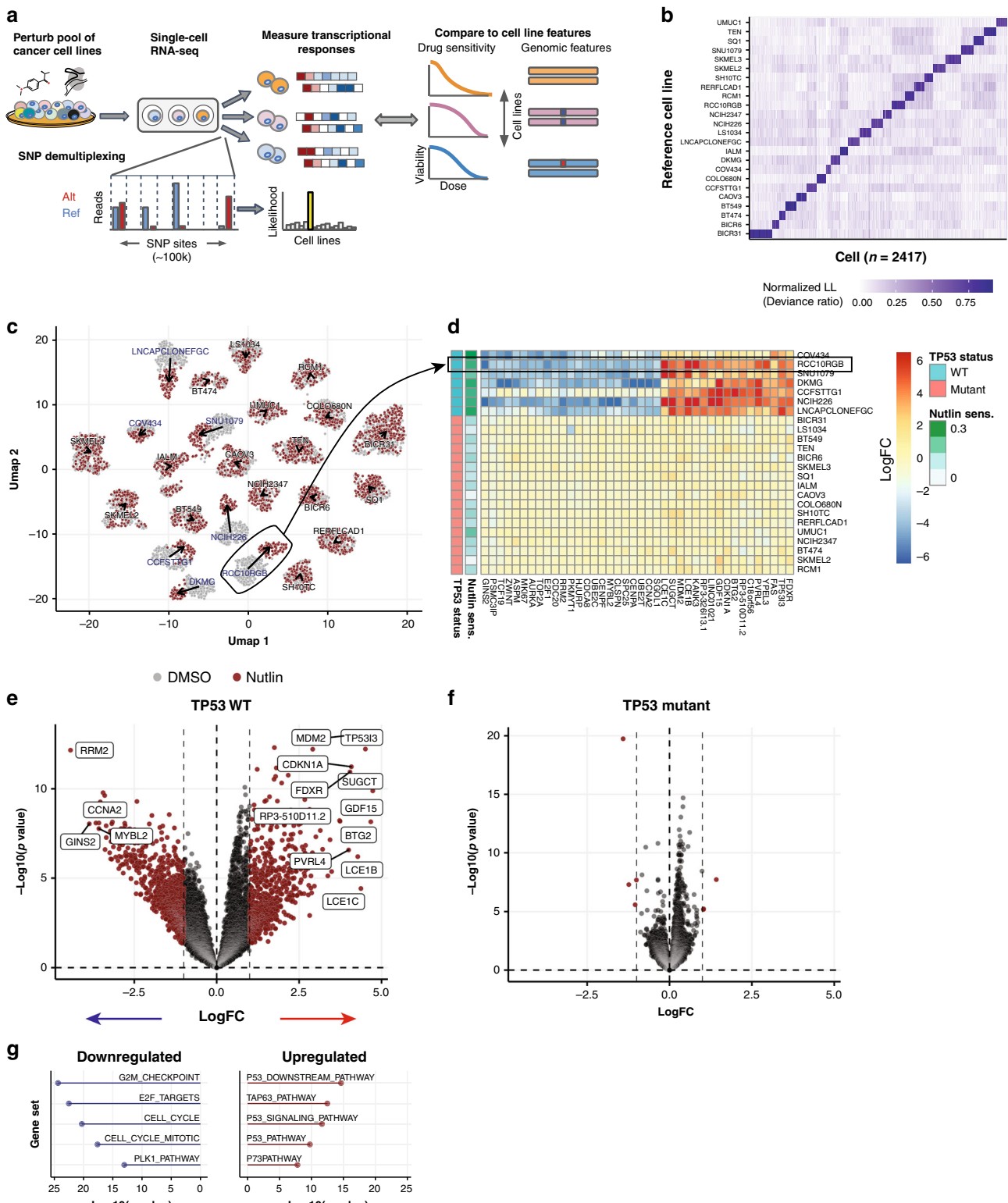

**Fig. 1 Multiplexed transcriptional profiling across pools of cell lines. a** Schematic diagram illustrating the MIX-Seq platform. **b** Heatmap showing likelihoods assigned by the SNP classification model for each cell coming from each parental cell line. The model consistently picks out a single cell line, from among the 24 "in-pool" cell lines, with high confidence. **c** UMAP representation of cells treated with DMSO control (gray) or nutlin (red) across a pool of 24 cell lines. Arrows indicate the shift in the population median coordinates for each cell line. **d** Heatmap showing average log fold-change estimates for each cell line for top differentially expressed genes. Nutlin sensitivity is given by 1 − area under dose response curve (AUC, see "Methods"). **e** Volcano plot showing strong gene expression changes in response to nutlin treatment across *TP53* WT cell lines ($n = 7$). Effect size estimates and *p* values (not corrected for multiple comparisons) for this and subsequent differential expression analyses are estimated using the limma-trend pipeline[49,50] ("Methods"). Vertical lines indicate a logFC threshold of 1. **f** Same as **e** for *TP53* mutant cell lines ($n = 17$), showing little gene expression change in response to nutlin treatment. **g** Gene set analysis identifies gene sets that are upregulated (right) and downregulated (left) by nutlin treatment in the *TP53* WT cell lines.

nutlin or vehicle control (dimethyl sulfoxide (DMSO)) in two-dimensional (2D) space revealed clear clustering by cell line, with shifts in the nutlin-treated cell populations for some cell lines but not in others (Fig. 1c). Estimates of the average drug-induced changes in gene expression for each cell line (see "Methods") revealed a robust response in each of the seven *TP53* WT cell lines in the pool, but only minimal changes in cell lines harboring *TP53* mutations, as expected (Fig. 1d–f). Furthermore, gene set enrichment analysis ("Methods") of the average transcriptional response among *TP53* WT cell lines showed clear upregulation of genes in the *TP53* downstream pathway, as well as down-regulation of cell cycle processes (Fig. 1g).

Across nearly all 13 drugs profiled, we were able to identify robust transcriptional response signatures, and these signatures were often highly informative about the compounds' MoA. For example, treatment with the proteasome inhibitor bortezomib elicited strong upregulation of protein folding and heat-shock response pathways (Supplementary Fig. 4a). The chemotherapy drug Gemcitabine altered the expression of apoptosis-related genes (Supplementary Fig. 4b), and mammalian target of rapamycin (mTOR) signaling was the top downregulated gene set following treatment with the mTOR inhibitor everolimus (Supplementary Fig. 4c). We also verified that transcriptional response profiles measured using MIX-Seq showed good overall agreement with those measured by the L1000 gene expression assay[10] for the same compounds (Supplementary Fig. 5). Taken together, these results demonstrate the ability of MIX-Seq to measure selective transcriptional effects of a drug across a pool of cell lines, and highlight the utility of such information for identifying a drug's cellular effects and MoA[10,12].

In addition to measuring drug-responses, MIX-Seq can also be used to study the transcriptional effects of genetic perturbations in cell line pools. As a proof of concept, we introduced two single-guide RNAs (sgRNAs) targeting the gene glutathione peroxidase 4 (*GPX4*) by lentiviral transduction into a pool of 50 cell lines, and performed scRNA-seq at either 72 or 96 h post-infection ("Methods"). There was robust on-target reduction of *GPX4* mRNA across all cell lines in the pool, and a transcriptional response consistent with the known function of *GPX4* in lipid metabolism (Supplementary Fig. 6).

**Deconvolution of viability-related response signatures**. A key advantage of profiling transcriptional responses across a large number of cell contexts using MIX-Seq is that it allows us to distinguish the overall transcriptional effects of a drug from the signature specifically associated with its viability effects. To do this, we employed a statistical modeling approach relating the transcriptional changes measured in each cell line to their via-bility response in the drug sensitivity data from GDSC and PRISM[4,17,20]. Specifically, we decomposed the change in expression of each gene into two components: a viability-independent response component ($\beta_0$) characterizing the response of com-pletely insensitive cell lines, and a viability-related response component ($\beta_1$) characterizing the difference between sensitive and insensitive cell lines ("Methods").

As an example, we first consider treatment of a 99 cell line pool with the MEK inhibitor trametinib, along with vehicle control (DMSO). We recovered more than 100 cells per cell line on average in each condition, detecting 97 of 99 cell lines with a minimum of 20 cells in each condition (average 130 cells/condition; Fig. 2a, b). Downsampling analysis suggested that measuring tens of cells per condition was sufficient to estimate each cell line's transcriptional response profile (Supplementary Fig. 7). The viability-independent response to trametinib included strong downregulation of mitogen-activated protein kinase

(MAPK) signaling genes, including *EGR1, ETV4/5, DUSP4/5/6,* and *SPRY2/4, KRAS* signaling pathways, and tumor necrosis factor-alpha signaling, as well as upregulation of the interferon response (Fig. 2c), consistent with previous reports[25,26]. In contrast, the viability-related component showed strong down-regulation of cell cycle processes (Fig. 2d), implicating a selective cell cycle arrest as mediating the long-term viability effects of trametinib. These results underscore how distinguishing the two response components can help understand a drug's MoA.

Applying this analysis across all eight compounds profiled with MIX-Seq that have selective viability effects, we found several core components of the viability-related response that were largely shared across compounds. These were highly enriched for cell cycle genes, which were selectively downregulated in the sensitive cell lines in virtually all the selective compounds profiled (Supplementary Fig. 8). Notably, the shared signature was also apparent in cells treated with broadly toxic compounds, such as prexasertib, the *BRD2*-inhibitor JQ-1, bortezomib, and gemcita-bine, suggesting it reflects a general transcriptional signature of decreased cell viability and/or proliferation. The two inhibitors of anti-apoptotic proteins—navitoclax and AZD5591—were unique among the compounds tested in that they did not produce robust transcriptional response signatures, despite eliciting strong selective viability responses (particularly in the case of AZD5591).

In order to determine how the number of different cell lines profiled impacts estimation of these transcriptional response components, we performed a downsampling analysis (Supple-mentary Fig. 9). While the average response across cell lines could be estimated reliably from relatively few (5–10) lines, estimates of the viability-related and viability-independent response compo-nents became more robust (as measured by their similarity to estimates using all cell lines) when including data from ≥50 lines (Supplementary Fig. 9).

**Prediction of long-term viability from MIX-Seq profiles**. The ability of MIX-Seq to efficiently profile transcriptional responses across many cell lines also allows us to test the feasibility of training models to predict the long-term viability effects of a drug from short-term transcriptional response measurements. Such an approach could have clinical applications in therapeutic response prediction, as patient cells can be transcriptionally profiled without long-term cultures. To test this, we used random forest models, assessing their accuracy using the $R^2$ of predictions on held-out test cell lines (tenfold cross-validation; "Methods").

Our models accurately predicted across-cell line differences in viability effects for virtually all drugs tested that had selective killing profiles (with the exception of the apoptosis-inducing compounds AZD5591 and navitoclax; Fig. 3a). For several drugs, the models could even predict viability responses from transcrip-tional changes measured just 6 h posttreatment. For comparison, we also trained models using the baseline "omics" features of the cell lines, including their baseline expression levels (from bulk RNA-seq data) and the presence of damaging and hotspot mutations[6,27]. Across drugs, we found that transcriptional response signatures were more predictive of long-term viability responses compared to the cell lines' baseline features (Fig. 3a; $n = 17$; $p = 8.4 \times 10^{-4}$; Wilcoxon signed-rank test). Notably, even when we used all available data to train models on the baseline features, such that they had access to much larger training samples (e.g., $n = 741$ vs. 24 cell lines for nutlin), transcriptional response profiles still compared favorably for predicting viability effects for most drugs (Supplementary Fig. 10).

We also successfully trained a single model ("Methods") to predict viability responses across all cell lines and drugs from transcriptional changes measured 24 h posttreatment with good

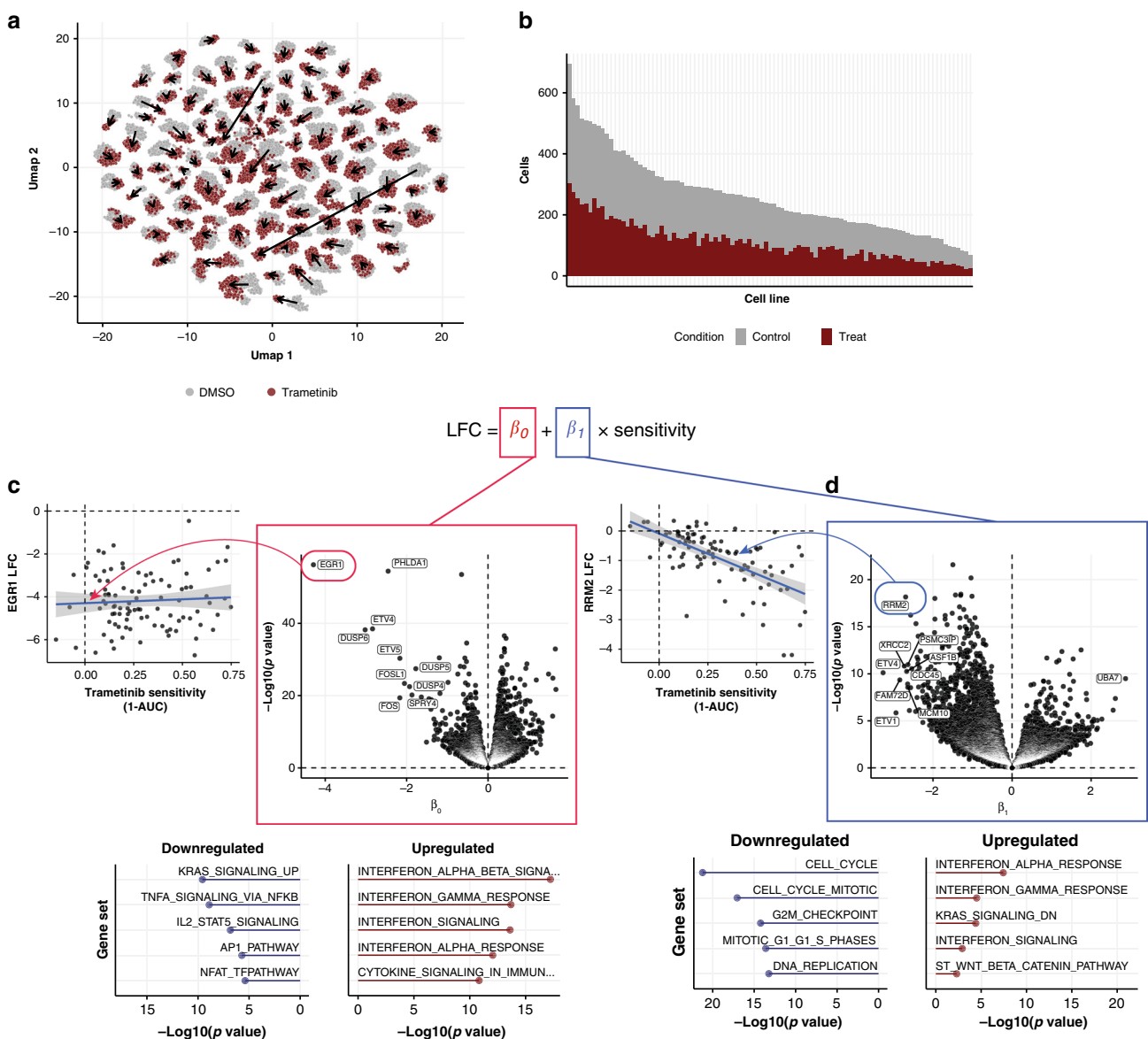

**Fig. 2 Viability-related and -independent transcriptional response components. a** UMAP representation of single-cell expression profiles in a 99 cell line pool treated with vehicle control (gray) or trametinib (red). Arrows indicate trametinib-induced shift of population median coordinates for each cell line. **b** Histogram showing the number of cells recovered in each cell line and condition. **c** Volcano plot showing the viability-independent response for each gene, representing the "y-intercept" of a linear fit of expression change to drug sensitivity (1-AUC; "Methods"). Inset at left shows this relationship for an example gene: *EGR1*. The blue line shows the linear regression trend line (with the 95% CI interval shown in shaded gray). Inset below shows top upregulated (right, red) and downregulated (left, blue) gene sets. **d** Same as **c** for the viability-related response.

accuracy ($R^2 = 0.50$), suggesting that there is a consistent transcriptional signature associated with viability effects across compounds. Furthermore, the genes whose transcriptional response most contributed to predicted viability effects were characterized by upregulation of *NFKB*, apoptosis, and *TP53* signaling, along with downregulation of translation, cell cycle, and *MYC* signaling (Fig. 3b), consistent with the previous analysis of viability-related response signatures (Supplementary Fig. 8). Together, these results suggest that posttreatment transcriptional signatures can provide a robust signal of cellular response to drugs that could be applied to predict their long-term viability effects.

Transcriptional profiling across large panels of cell lines also enables identification of the factors underlying their variable drug responses without a priori knowledge of the relevant genomic/molecular features driving such differences. As a simple illustration of this, we applied principal component analysis (PCA) to the

matrix of trametinib responses across cell lines, measured 24 h posttreatment (Fig. 3c). The first principal component (PC1), captured differences in trametinib sensitivity across cell lines (Fig. 3d, e). Indeed, across 9/13 tested drugs, PC1 or PC2 of the transcriptional response matrix (measured at 24 h posttreatment) was significantly correlated with the cell lines' measured drug sensitivity (false discovery rate (FDR) < 0.1; Supplementary Fig. 11), suggesting that this is often a predominant source of response heterogeneity. For trametinib, PC2 identified a pattern of differential response among trametinib-sensitive cell lines, distinguishing the responses of mostly *BRAF* mutant melanoma lines from other sensitive lines (largely KRAS mutant) (Fig. 3f). These first two PCs were also recapitulated in a separate experiment measuring trametinib responses in a different pool of cell lines (Supplementary Fig. 12). This example thus highlights the power of transcriptional profiling across cell contexts to identify multiple

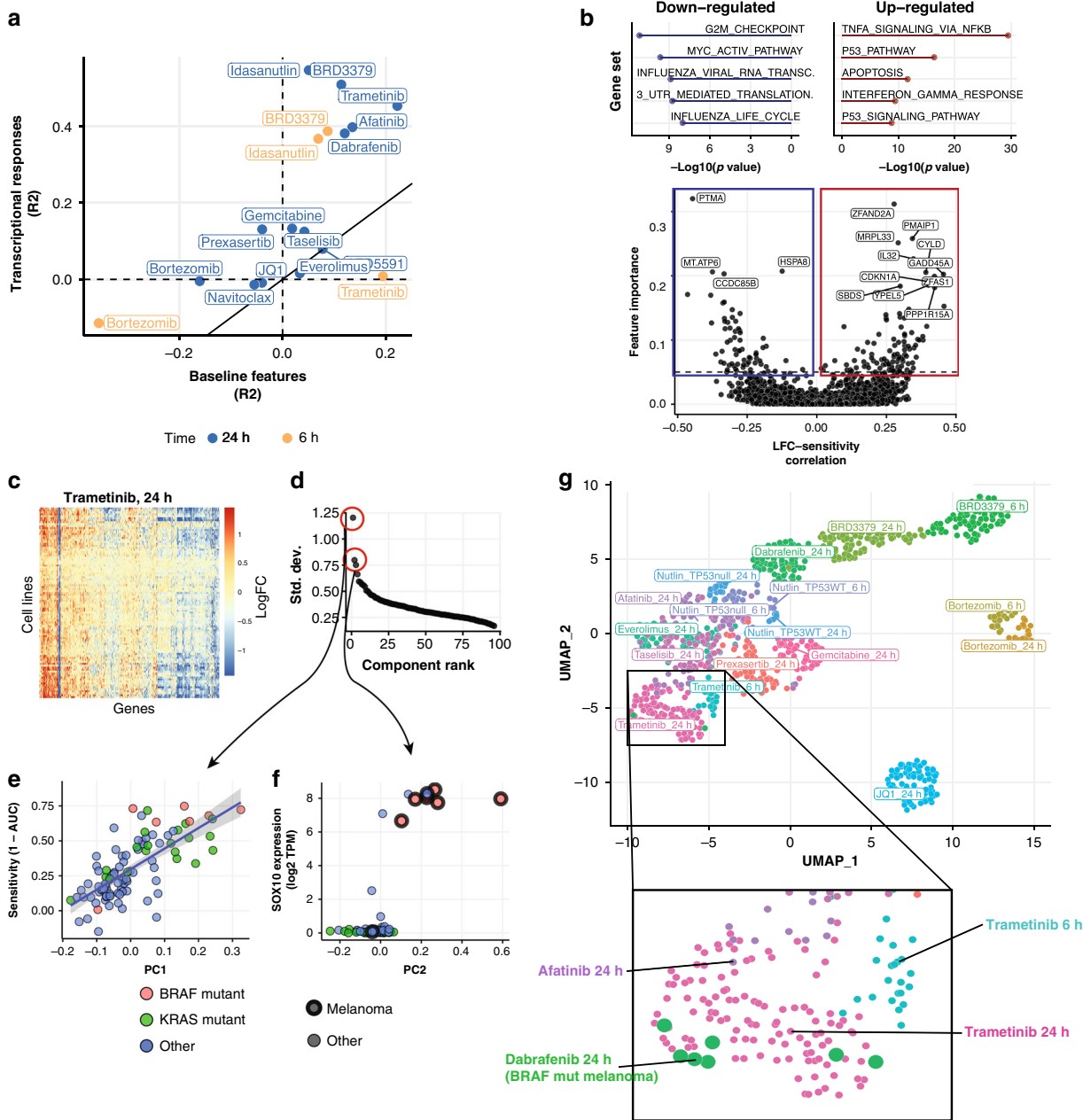

**Fig. 3 Machine learning analysis powered by large-scale transcriptional profiling. a** Accuracy of models trained to predict sensitivity for each individual drug, using either measured transcriptional responses or baseline "omics" features of the cell lines (using the same set of cell lines). Predictions based on transcriptional profiling at 6 and 24 h posttreatment are indicated by the gold and blue dots, respectively. **b** Plot of the Pearson correlation between each gene's transcriptional response (24 h posttreatment) and drug sensitivity (1 − AUC) across cell lines and drugs, compared with its feature importance for the random forest predictive models. Gene set enrichment analysis of highlighted genes shown above. **c** Matrix of measured transcriptional responses across cell lines 24 h after trametinib treatment. **d** Eigenvalue spectrum of PCA applied to the matrix from **c**. **e** The projection of each cell line's response onto PC1 is plotted against its measured trametinib sensitivity (red and green points indicate cell lines with activating BRAF or KRAS mutations respectively). Linear regression trend line (with the shaded gray region showing the 95% CI interval) is shown in blue. **f** Comparison of PC2 scores with expression of the melanoma-specific transcription factor *SOX10* across cell lines. **g** UMAP embedding of transcriptional response profiles across drugs, cell lines, and posttreatment time points. Points are colored by treatment condition (drug and time point). Inset below shows zoomed view of the region indicated by the rectangle, with the larger green dots representing responses of *BRAF* mutant melanoma lines to the BRAFi dabrafenib.

biologically relevant factors underlying the differential cellular response to the drug.

Finally, in order to identify, in an unsupervised manner, global patterns of transcriptional responses across cell lines, compounds, and time points, we created a 2D embedding of all the combined perturbation response profiles with UMAP[28]. While perturbation response profiles mostly grouped by perturbation type (drug and posttreatment time point) (Fig. 3g), relationships between the set of responses for related perturbation types were also apparent. For example, responses to the same drug profiled at multiple posttreatment time points were nearby in UMAP space, and functionally related drugs such as taselisib (PIK3CAi) and everolimus (MTORi), as well as trametinib (MEKi) and afatinib (EGFRi), clustered near one another. Interestingly, the response of *BRAF* mutant cell lines to

the BRAFi dabrafenib grouped with trametinib response profiles, rather than with the other dabrafenib responses (Fig. 3g).

**Single-cell resolution uncovers heterogeneous responses.** In addition to the benefits of scRNA-seq as a tool for efficiently multiplexing transcriptional profiling across many cell lines, the ability to characterize responses with single-cell resolution enables qualitatively new analyses not possible with bulk RNA-seq, across diverse phenotypes, all assessed simultaneously with a single assay.

For example, characterizing the cell cycle effects of a perturbation can yield important information about its MoA. Such measurements are typically made using a fluorescence-activated cell sorting-based assay, which must be performed on each sample independently and cannot be used to relate cell cycle effects post hoc to other phenotypes without advanced planning. Using MIX-Seq, we can make such measurements in parallel across a pool of cell lines, inferring the cell cycle phase of each cell from its transcriptional profile[29]. As a demonstration, we applied this approach to the nutlin treatment experiment (Fig. 1), finding that nutlin elicited a pronounced G0/G1-arrest phenotype selectively among the TP53 WT cell lines (Fig. 4a, b), as expected.

Next, we systematically assessed the effects of each compound on the cell cycle ("Methods"). At 24 h posttreatment, most drugs produced an increase in the proportion of cells in G0/G1 (10/13 drugs) and concomitantly decreased the proportion of cells in S (9/13) and G2/M phases (9/13), consistent with cell cycle arrest at the G1/S transition (Fig. 4c). Two notable exceptions were the DNA-damaging agent gemcitabine and the CHEK1/2 inhibitor prexasertib. Gemcitabine also decreased the proportion of cells in G2/M but with an increase in S-phase cells, consistent with its known role in triggering CHEK1-mediated S-phase arrest. Prexasertib decreased the proportion of S-phase cells and slightly increased the fraction of G2/M cells, consistent with inhibition of CHEK1-mediated DNA-damage checkpoints leading to disregu-lated progression of cells through the cell cycle[30].

For selective compounds, cell cycle effects were also well correlated with measured viability effects, such that drugs typically had larger effects in more sensitive cell lines (Supple-mentary Fig. 13). We also used the single-cell profiles to directly estimate drug-induced changes in relative cell abundance, finding that selective compounds consistently decreased the representa-tion of more sensitive cell lines in the pool, particularly when measured 24 h posttreatment (Supplementary Fig. 14). Notably, this relationship was observed most strongly with the MCL1 inhibitor AZD5591, despite the fact that we did not observe a robust transcriptional response to the drug, suggesting that direct induction of apoptosis may be detectable by selective cell line dropout in the absence of marked transcriptional changes. Together, these results demonstrate that MIX-Seq can reliably read out the effects of perturbations on cell cycle progression as well as overall cell viability.

Unlike bulk expression profiling, scRNA-seq also enables the characterization of heterogeneous responses across the cells in a population. For example, bortezomib treatment elicited a bimodal response for 10 of the 24 cell lines in the pool (Fig. 4d, e). All ten cases showed a similar pattern, with one cell subset arresting in G0/G1 and another composed of predominantly S-phase cells (Fig. 4e, f). In contrast, we observed more homogenous population responses for the other drugs tested.

When cell lines are composed of transcriptionally distinct subsets even in the absence of perturbations, as we showed in a recent study[31], MIX-Seq can be used to examine whether different cell populations within a given sample exhibit differential treatment responses. For example, the lung cell line IALM had two distinct subpopulations at baseline, characterized by differential expression of epithelial-to-mesenchymal transition and integrin-related pro-grams (Supplementary Fig. 15). Indeed, these subpopulations exhibited subtle but significant differences in their response to trametinib treatment (Supplementary Fig. 15). These examples highlight the ability of MIX-Seq to reveal heterogeneous responses that would be missed by bulk transcriptional profiling.

**Multiplexed profiling across posttreatment time points.** Many perturbations elicit cellular responses that evolve over time, suggesting that more information could be obtained by profiling cells across a sequence of posttreatment time points. Several methods have recently been developed for introducing sample-specific barcodes to allow multiplexing of scRNA-seq measure-ments across experimental conditions and time points[23,32]. In particular, Cell Hashing[23] uses oligonucleotide-conjugated anti-bodies against cell-surface antigens (called hashtags) to label cells with unique barcodes for each experimental condition. Since MIX-Seq uses naturally occurring SNP barcodes to multiplex cell lines, it can easily be combined with such approaches to allow for dual multiplexing of cell lines and experimental conditions with a single scRNA-seq readout.

Leveraging this, we measured responses of a pool of 24 cell lines to trametinib along 5 time points, ranging from 3 to 48 h posttreatment, using Cell Hashing to multiplex treatment conditions (Fig. 5a). As controls, we included DMSO-treated samples at each of the 5 time points, in addition to untreated samples, for a total of 11 conditions. Hashtag reads provided robust labeling of treatment conditions, with good tagging efficiency across all cell lines (Supplementary Fig. 16). Since we did not observe substantial differences in DMSO-treated cells across time points (Supplementary Fig. 17), we pooled them together for subsequent analysis, yielding a total of 13,713 clearly tagged single cells across all treatment conditions and cell lines.

The single-cell expression profiles illustrated strong time-dependent changes in response to trametinib, whose magnitude varied considerably across cell lines (Fig. 5b). To better under-stand these changes, we examined the temporal transcriptional changes of key trametinib-response genes. For example, EGR1, an immediate early response gene known to be activated by MAPK signaling[33], was dramatically downregulated 3 h after trametinib treatment in both the sensitive cell line RCM1 and the insensitive line TEN (Fig. 5c). In contrast, MCM7, a cell-cycle-related gene that was part of the viability-related response, was selectively downregulated only in the sensitive line RCM1 and only after 12–24 h posttreatment (Fig. 5d).

We next applied our statistical model (Fig. 2c, d) to quantify the temporal evolution of viability-related and viability-independent components of the trametinib response for each gene, integrating across all cell lines. Downregulated genes in the viability-independent response showed a range of temporal patterns, with several (such as EGR1 and DUSP6) reaching maximal down-regulation 3 h posttreatment (Fig. 5e). In contrast, the viability-related response emerged much later, with genes such as CLSPN, ESCO2, and NCAPG showing selective downregulation in sensitive cell lines only 12–24 h posttreatment (Fig. 5f). We confirmed that these results were not biased by temporal variation in the numbers of cells available to estimate each cell line's response (Supplemen-tary Fig. 18). We also characterized these differences at the pathway level, finding that the viability-independent downregulation of the KRAS signaling pathway emerged 3 h after treatment (Fig. 5e), while the viability-related downregulation of cell cycle genes started 24 h after treatment (Fig. 5f). The latter was also consistent with the time course of G0/G1 arrest based on inferred cell cycle phases (Fig. 5g).

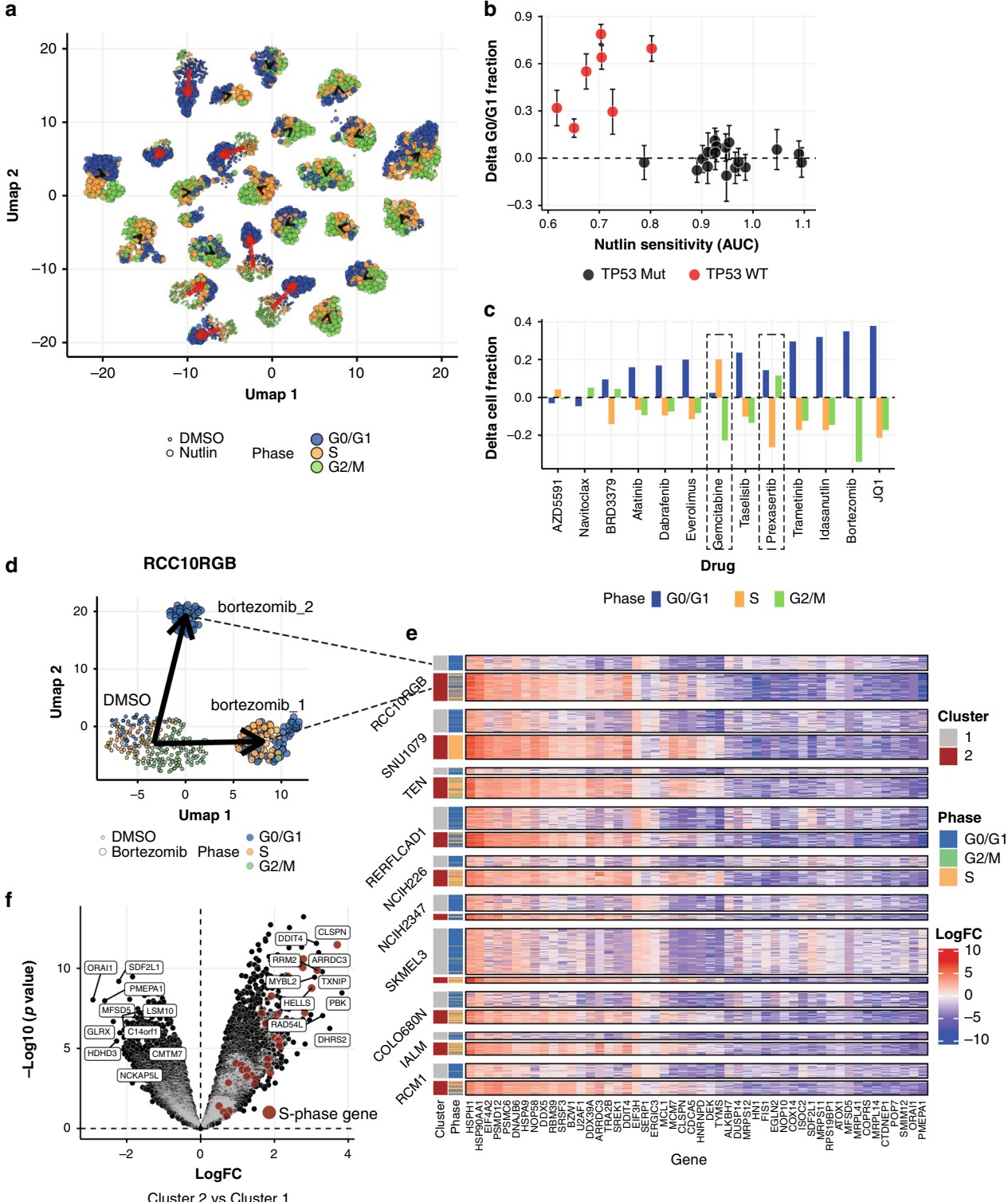

**Fig. 4 Population heterogeneity in pre- and post-perturbation transcriptional programs. a** UMAP representation of gene expression profiles of DMSO (smaller points) and nutlin-treated (larger points) cell populations for a pool of 24 cell lines (as in Fig. 1). Cells are colored by their inferred cell cycle phase. *TP53* WT cell lines (red arrows) show predominance of G0/G1-phase cells after nutlin treatment not observed in *TP53* mutant cell lines (black arrows). **b** Quantification of the change in proportion of cells in G0/G1 in each cell line (*y*-axis) shows that nutlin treatment elicits G1 arrest selectively among the *TP53* WT cell lines. Error bars show the 95% CI for estimates of the change in G0/G1 cell proportions. **c** Average change in the fraction of cells in each cell cycle phase for each drug treatment (averages are weighted by measured drug sensitivity, all for 24 h time points). **d** UMAP plot showing two subpopulations of RCC10RGB cells emerging 24 h after treatment with bortezomib. Dot color depicts inferred cell cycle phase and dot size represents treatment condition. **e** Heatmap showing LFC relative to control cells of top differentially expressed genes between the two clusters of bortezomib-treated cells for each of the 10/24 cell lines that showed a bimodal response pattern. In nearly all cases, one of the clusters (labeled cluster 2 by convention) was characterized by a predominance of S-phase cells, compared with mostly G0/G1 cells in cluster 1. **f** Volcano plot showing differentially expressed genes in cluster 2 vs cluster 1 averaged across the nine cell lines depicted in **e**. Genes that are part of the S-phase signature are highlighted in red.

These results thus highlight the utility of large-scale transcriptional profiling, both across cell lines and time points, to identify the different components of drug response. The ability to separate these transcriptional components could provide clues to both the initial effects of target engagement, as well as the mechanism underlying selective loss of cell viability, and more powerfully inform MoA.

## Discussion

Here we present an experimental and computational platform (MIX-Seq) for performing highly multiplexed transcriptional profiling of perturbation responses across many cell contexts using scRNA-seq applied to co-treated pools of cancer cell lines. We demonstrate this approach by profiling the responses of pools of 24–99 cell lines to a range of different drugs, as well as to CRISPR perturbations.

To determine the cell line identity of each cell, we developed an optimized computational demultiplexing method that utilizes their unique SNP profiles, showing that it is able to classify single cells with negligible error rates, even at low sequencing depths. This approach also allows for identification of droplets containing ambient mRNA (empty droplets) or two cells ("doublets"), Our method is similar to other recently published SNP-demultiplexing methods[34,35], most notably Demuxlet[22] that also uses pre-computed reference SNP profiles. However, rather than detecting doublets by explicitly computing the likelihood of all possible reference mixtures, as in Demuxlet, we utilized a fast approximation based on generalized linear models (GLMs; "Methods") that efficiently scales to much larger cell line pools. For smaller pools where both models could be applied, we verified that our model produced identical single-cell classifications and similar doublet detection results compared to Demuxlet (Supplementary Fig. 1).

A number of approaches have been developed for high-throughput transcriptional profiling that can be used to study perturbation responses at scale. The CMAP project has utilized a low-cost bead-based assay that measures a reduced set of ~1000 "landmark" genes to profile thousands of different perturbation responses[10,12]. More recently, methods such as DRUG-seq[15] and PLATE-seq[14] use oligo-tagging of treatment conditions to perform multiplexed RNA-sequencing, greatly reducing library preparation costs. Similar sample-barcoding strategies have also been employed with scRNA-seq[21,23,32], allowing for multiplexed profiling across treatment conditions such as time points and drugs. MIX-Seq complements these existing approaches by allowing for multiplexed profiling of perturbation responses across broad panels of heterogeneous cell contexts, without the need for additional experimental barcoding steps. As we demonstrate, MIX-Seq can also be combined with existing sample-barcoding strategies, such as Cell Hashing, to enable dual multiplexing across treatment conditions and cell contexts.

The single-cell resolution of MIX-Seq also enables a more detailed characterization of the perturbation responses of cancer cell populations. For example, we show that analysis of changes in the inferred cell cycle composition across cell lines can provide insights into the mechanisms underlying decreased proliferation. We also demonstrate examples of how single-cell profiling can be used to reveal heterogeneous responses within a sample, as well as to isolate differential responses among distinct subpopulations. Such capabilities could be particularly important when studying more heterogeneous samples such as primary tumor samples, as well as for probing mechanisms of drug resistance.

MIX-Seq's ability to efficiently profile genome-wide transcriptional responses across a broad panel of cell lines provides several advantages relative to traditional approaches. First, it allows for the detection of context-specific responses, which could

be critical for highly selective drugs like dabrafenib and nutlin. Even among sensitive cell lines, however, there can be substantial response heterogeneity, and profiling many cell lines naturally makes results less sensitive to the particular choice of cell line models under study. For example, we found that responses to the *BRAF*-inhibitor dabrafenib showed substantial variation, even among the highly sensitive *BRAF*-mutant melanoma cell lines (Supplementary Fig. 19).

By profiling perturbation responses across large panels of well-characterized cell lines, we can also uncover how patterns of transcriptional changes relate to the underlying genomic and functional features of the cells. In particular, pairing MIX-Seq with PRISM[20], which can measure long-term drug sensitivity across the same panel of cell lines, allows us to dissect the components of transcriptional response associated with decreased cell viability in order to better understand the mechanisms underlying a drug's fitness effects. For the drugs studied here, we found that viability-related responses were broadly similar across drugs, mostly reflecting a downregulation of cell cycle genes and upregulation of genes involved in translation (Supplementary Fig. 8), though transcriptional signatures associated with apoptosis were also observed for some drugs (e.g., Supplementary Fig. 4). The two clear exceptions to this pattern were both inhibitors of anti-apoptotic proteins—the BCL-2i navitoclax and the MCL1i AZD5591. These drugs did not produce strong and/or selective transcriptional responses. This suggests that compounds which directly induce apoptosis may not elicit a clear transcriptional signature, at least when measured 24 h posttreatment as done here.

One potential caveat of profiling transcriptional responses in pools of cell lines is that paracrine signaling between cell lines in the pool could affect the measured responses. We found that scRNA-seq profiles at baseline for cells grown in a pooled format were consistently most similar to bulk RNA-seq measurements of the same cell lines grown individually (Supplementary Fig. 2), suggesting that such paracrine signaling effects are likely to be modest. Measuring treatment and control conditions within the same pool of cell lines also provides some internal control for baseline effects of paracrine signaling. Finally, previous work has shown that drug response profiles measured in cell line pools are largely concordant with standard measurements[20]. Nevertheless, the potential for interactions between cell lines in the pool must be considered when measuring perturbations responses using MIX-Seq.

We also used MIX-Seq to show that transcriptional responses measured 6–24 h after drug treatment can be used to predict long-term cell viability remarkably well across selected targeted cancer drugs. These results are in broad agreement with recently published analyses[36,37] comparing drug sensitivity data and transcriptional profiling data from the CMAP project[10,12], spanning many compounds in a core set of cell lines. By allowing efficient profiling of a given drug's transcriptional effects across many cell lines, MIX-Seq offers unique opportunities to evaluate these relationships in detail for individual compounds, rather than requiring analyses that pool data across compounds, as in previous work. Notably, for the drugs tested here, the ability of machine learning models to predict drug sensitivity from a cell line's transcriptional response was substantially better than when using baseline omics features. These results suggest that transcriptional profiling could be used as a robust pharmacodynamic marker of drug sensitivity, which may provide improved predictions of tumor vulnerabilities compared with standard biomarker approaches. An important potential future application of this approach would be to utilize scRNA-seq to rapidly assess the sensitivity of primary tumor cells to various drug treatments ex vivo, circumventing the prolonged primary cell cultures needed to achieve sufficient cell numbers for standard long-term

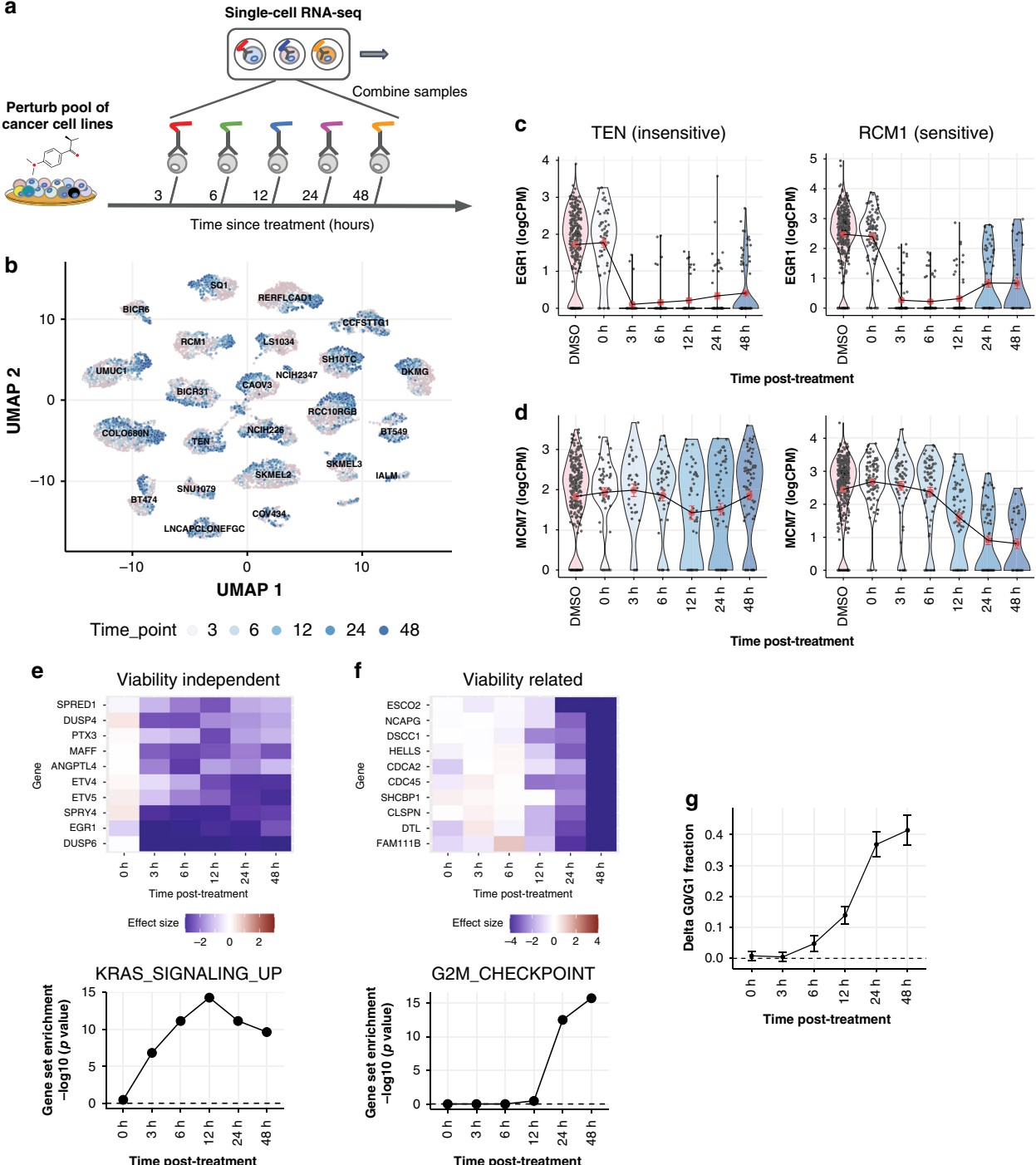

**Fig. 5 Dual-multiplexed transcriptional profiling across cell lines and time points. a** Schematic diagram illustrating experiment using Cell Hashing to multiplex scRNA-seq of cell line pools sampled at different time points following drug treatment. **b** UMAP plot showing 13,713 cells across a pool of 24 cell lines at different times following treatment with trametinib (shades of blue) or DMSO control (pink). **c** Single-cell expression levels of *EGR1* at different time points following trametinib treatment for an example insensitive/sensitive cell line (left/right). Red dots depict the mean expression levels at each time point, and error bars show the interval $+/-$ s.e.m. **d** Same as **c**, for *MCM7*. **e** (Top) Time course of the viability-independent response for top downregulated genes. (Bottom) Enrichment of HALLMARK_KRAS_SIGNALING_UP genes in the downregulated viability-independent response at each time point. **f** (Top) Same as **e**, showing time course of the viability-related response for top downregulated genes. (Bottom) Enrichment of HALLMARK_G2M_CHECKPOINT genes in the viability-related response at each time posttreatment. **g** Average time course of G0/G1 arrest across cell lines ($n = 24$ cell lines). Error bars indicate interval $+/-$ s.e.m.

viability assays, such as Cell-Titer-Glo[38,39]. However, it will be important to extend these tests to a broader range of drugs and primary patient-derived cancer models, in order to understand the generalizability of these results.

We envision that MIX-Seq could be used to efficiently build a database of transcriptomic changes elicited by a broad range of different chemical and genetic perturbations, each measured across a large heterogeneous panel of cancer models. Analogously

to the CMAP project[10,12], such a database could be used for predicting the mode of action of compounds and genetic manipulations whose cellular effects remain to be uncovered. By measuring perturbation responses across many different cell contexts, with single-cell resolution, MIX-Seq provides a powerful tool for identifying the core transcriptional programs of cancer cells and better understanding how perturbations interact with the underlying cell context to alter these programs.

## Methods

**Method of cell line pooling.** Cell line pools were made in sets of 25 cell lines. These 25 cell line pools were chosen based on doubling time and were grown in RPMI without phenol red and with 10% fetal bovine serum (FBS). Cell lines were then washed with 10 mL of phosphate-buffered saline (PBS) and trypsinized with 1 mL trypsin, which was then removed. Ten milliliters of RPMI media were added to the cells post trypsinization and resuspended. Cells were then counted by a Nexcelom cellometer using 10 μL of cell suspension and 10 μL of Trypan blue. Equal numbers of cells per cell line were mixed together and spun down at 1250 rpm for 5 min. Media was aspirated and the cells were resuspended in Sigma Cell Freezing media and frozen in 1 mL aliquots. This process was repeated for all of the 25 cell line pools. For MIX-Seq experiments involving larger pools, multiple 25 cell line pools were thawed in RPMI with 10% FBS, spun down, and resuspended in 5 mL of RPMI media. Cells were then counted and equal numbers were combined together on the day of plating to form larger pools of up to ~100 cell lines.

**Cell culture.** For drug treatment experiments, cell line pools were cultured in RPMI containing 10% FBS but did not contain phenol red or penicillin/streptomycin. Cell line pools were validated as mycoplasma free prior to initiating the experiment. Cell line pools were plated at 200,000 cells per well in 6-well plates containing 2 mL of RPMI culture media described above. Cell seeding density did not vary depending on pool size (~25, 50, or 100 cell line pools). Cell pools were plated ~16–20 h prior to drug treatment. Cells were treated with the described drugs or vehicle (DMSO) with a 0.2% final media DMSO concentration.

For GPX4 knockout, cell line pools were plated at 200,000 cells per well in 12-well plates containing 1 mL of RPMI culture media. Twenty-four hours later, the cells were infected with lentivirus expressing Cas9 and sgRNA at a multiplicity of infection of 20 in the presence of 4 μg/mL of polybrene. At 48 h after the infection, the culture medium was replaced with medium containing 1 μg/mL puromycin. Cells were harvested at 72 or 96 h after the infection.

**Cell harvesting.** Generally cells were harvested after drug treatment using standard cell culture methods. After drug treatment, cells that were in suspension (presumably containing dead cells from drug treatment) were collected and reserved for addition to the adherent cell fraction. Adherent cells were washed once with 1× PBS, trypsinized in 1 mL trypsin, incubated for 3–7 min at 37 °C, and then trypsin inactivated with 1 mL growth media. For Cell Hashing, cells were treated with TrypLE Express (ThermoFisher) instead of trypsin to reduce the amount of cell surface proteins digested that may affect the binding of Cell Hashing antibodies.

For the trametinib time course experiment, cells were treated with trametinib with a staggered dosing schedule so all time points could be collected simultaneously. Cells were plated 19 h prior to the first drug treatment, corresponding to the 48 h time point. Cells were harvested for 10x capture ~67 h after initial seeding. Final concentrations for drug treatments are listed in Supplementary Data 3.

**Preparation of cell suspensions and scRNA-Seq.** After trypsinization, adherent and suspension cells were combined for each treatment, pelleted, and resuspended in Cell Capture Buffer (1× PBS with 0.04% bovine serum albumin (BSA)). Cells were counted (including Trypan blue non-viable cells) and resuspended at a concentration of 1000 cells per microliter for standard loading on the Chromium Controller (10x Genomics) or at 1500 cells per microliter for "super loaded" samples. Up to 40,000 cells were loaded per 10x channel for "super loaded" samples, with expected recovery of up to 20,000 cells per channel. Cell suspensions were captured on a 10x Chromium controller using Single Cell 3' reagent chemistries (either version 2 or version 3 reagents) (Supplementary Data 2).

**Cell Hashing cell labeling.** Cell Hashing[23] was performed using the cell harvest method described above with the following changes. All steps were performed on ice. Harvested cells were resuspended in Cell Hashing Staining Buffer (1× PBS with 2% BSA and 0.02% Tween) prior to cell counting. Samples were counted in duplicate with two technical replicates by Countess (Life Technologies) to estimate total cell number. Up to 1,000,000 cells (range 3e5–1e6 cells) were resuspended in 100 μL of Cell Hashing Staining Buffer. Cells were blocked with 10 μL of Human TruStain FcX blocking solution (BioLegend) for 10 min at 4 °C. One hundred-microliter cell suspensions in Cell Hashing Staining Buffer were then incubated with 2 μL of the appropriate BioLegend TotalSeq™-A Hashing antibody (a 1:50 dilution, using a total of 1 μg of antibody per cell suspension). TotalSeq™-A anti-human Hashtag antibodies #4-10 and #12-15 (product codes: 394607,

394609, 394611, 394613, 394615, 394617, 394619, 394623, 394625, 394627, 394629) were used. Cells were washed three times with 0.5 mL of Cell Hashing Staining Buffer and filtered through low-volume 40-μm cell strainers (Flowmi). All cell suspensions were recounted to achieve a uniform concentration of 1500 cells per microliter before pooling for 10× capture. For a detailed general protocol, please see our protocols.io at https://www.protocols.io/private/4AC0F6594480B498D8B60EAF6F518E66.

**Cell Hashing library preparation.** Separation of hashtag oligo (HTO)-derived cDNAs (<180 bp) and mRNA-derived cDNAs (>300 bp) was done after whole-transcriptome amplification by performing 0.6× SPRI bead purification (Agencourt) on cDNA reactions as described in 10x Genomics protocol. Briefly, the supernatant from 0.6× SPRI purification contains the HTO fraction, which was subsequently purified using two 2× SPRI purifications per the manufacturer's protocol (Agencourt). HTOs were eluted by resuspending SPRI beads in 15 μL TE.

Purified HTO sequencing libraries were then amplified by PCR. PCR reactions are given in Table 1.

Typically three identical "dial out" PCR reactions were performed per HTO library. We varied the number of PCR cycles to avoid under or overamplifying the HTO libraries. PCR cycling conditions are given in Table 2.

PCR reactions were purified using another 2× SPRI clean up and eluted in 15 μL of 1× TE. HTO libraries were then analyzed for amplification quality. Libraries were quantified by Qubit High sensitivity DNA assay (ThermoFisher) and loaded onto a BioAnalyzer high sensitivity DNA chip (Agilent) to determine whether an intended HTO product size of ~180 bp was achieved.

**Sequencing.** Samples were sequenced using HiSeq X (Illumina) or NovaSeq 6000 (Illumina) platforms. The read structure used (for 10 × 3' v3 chemistry) is given in Table 3.

The hashing library for the trametinib time course experiment was sequenced twice with spike-ins of 2.5–10%.

**Data processing.** Sequencing data were processed using the 10x Cell Ranger software, run using the Cumulus cloud-based analysis framework[40]. Our initial

---

**Table 1 PCR conditions for HTO library amplification.**

| Reagent | Volume |
| --- | --- |
| Purified HTO fraction after 2× SPRI | ~1 μL (5 ng) |
| 2× NEB Next Master Mix | 25 μL |
| Illumina TruSeq DNA D7xx_s primer (containing i7 index) 10 μM | 1 μL |
| SI PCR oligo 10 μM | 1 μL |
| H₂O | To 50 μL final volume |

---

**Table 2 PCR cycling conditions for HTO library amplification.**

| Step | Temperature | Time |
| --- | --- | --- |
| 1 | 98 °C | 10 s |
| 2ᵃ | 98 °C | 2 s |
| 3ᵃ | 72 °C | 15 s |
| 4 | 72 °C | 1 min |

ᵃSteps 2 and 3 were repeated for 15, 18, or 22 cycles.

---

**Table 3 Read structure used for sequencing.**

| Platform | Read | Cycles |
| --- | --- | --- |
| HiSeq | Read 1 | 28 (26)ᵃ |
| | Read 2 | 96 |
| | Index 1 | 8 |
| NovaSeq | Read 1 | 28 (26)ᵃ |
| | Read 2 | 80 |
| | Index 1 | 8 |

ᵃFor 10x 3' v2 chemistry.

experiments were done with 10x Single Cell 3' v2 chemistry and were processed using version 2 of the Cell Ranger software. In our later experiments, we used v3 chemistry and the corresponding version 3 of Cell Ranger. Reads were aligned to the hg19 reference genome.

**SNP identification**. To define a SNP panel for cell line classification, we identified SNPs that occurred frequently across a large panel of 1160 cell lines and that were also detected in scRNA-seq data. Specifically, we ran MuTect 1 (version 1.1.6) to call SNVs from bulk RNA-seq data and scRNA-seq data from 200 cell lines using a downsample to a coverage rate of 1000 and a fraction contamination rate of 0.02 and with all other parameters set to defaults. We took the subset of SNPs that were observed in both the bulk and single-cell data, then ordered all SNPs by the frequency of their occurrence (in the bulk RNA-seq data), selecting the 100,000 most frequently observed SNPs.

For the bulk RNA-seq data, we used Freebayes[41] to estimate allelic fractions across the reference SNP panel, using the settings "pooled-continuous" and "report-monomorphic," and adding a pseudo-count of 1 to the reference and alternate allele read counts. For the single-cell data, we used the method scAlleleCount (https://github.com/barkasn/scAlleleCount) to extract reference and alternate allele counts at all SNP sites.

**SNP-based cell line classification**. To estimate the likelihood of the observed SNP reads for an individual cell having come from each reference parental cell line, we use a GLM approach. Specifically, we use a logistic regression model, where the probability of a read at SNP site $i$ being an alternate allele is given by:

$$\pi_i = \sigma(\beta_0 + \beta_j X_{ij}), \tag{1}$$

where $\sigma$ is the logistic function, $X_{ij}$ represents the (predefined) fraction of reads at SNP site $i$ from the alternate allele in cell line $j$ (estimated from bulk RNA-seq data), and $\beta$ are parameters estimated for each single cell and reference cell line by maximizing the likelihood:

$$\prod_i \mathcal{L}(\beta | y_i, n_i), \tag{2}$$

where $\mathcal{L}$ is the binomial likelihood, $y_i$ is the number of alternate allele reads, and $n_i$ is the total number of reads observed at site $i$. We fit models using the glm function in R, and the cell line whose SNP allelic fraction profile produced the highest likelihood for the observed single-cell SNP reads was selected. Goodness of fit was quantified by the deviance ratio: $1 - \text{deviance\_fit}/(\text{deviance\_null})$. We also compute a measure of the classification confidence, given by the margin between the best-fitting and second-best-fitting model deviance ratios, normalized by the standard deviation of deviance ratio values across reference cell lines (excluding the best matching cell line $j*$).

Estimates of the SNP classification error rate were given by $\frac{n_{out}}{n_{tot}} c$, where $n_{out}$ is the number of cells erroneously classified as out-of-pool cell lines, $n_{tot}$ is the total number of cells recovered in the experiment (excluding doublets and low-quality cells), and $c$ is a correction factor to account for the probability of cells being classified incorrectly among the in-pool cell lines. Assuming errors are made with equal probability among all reference cell lines, this is given by:

$$c = 1 - \frac{N_{pool} - 1}{N_{ref}}. \tag{3}$$

**Modeling doublets**. Doublet detection is performed using a similar GLM approach, where alternate allele probabilities are modeled as a mixture of the allelic fraction profiles from two reference cell lines $X_j$ and $X_k$:

$$\pi_i = \sigma(\beta_0 + \beta_j X_{ij} + \beta_k X_{ik}), \tag{4}$$

where the ratio $\beta_j/\beta_k$ represents the proportion of mRNA reads from cell line $j$ vs cell line $k$. In order to efficiently estimate the most likely pairwise mixture of reference cell lines, we use a Lasso-regularized GLM (implemented in the R package glmnet[42]), considering the allelic fraction profiles for all in-pool reference cell lines $X_j$ as covariates. We constrained coefficient estimates to be non-negative and limiting the model to use a maximum of two non-zero coefficients (i.e., two reference SNP profiles). After using the Lasso model to estimate the most likely doublet pair of cell lines, we then refit the GLM without regularization to estimate the goodness of fit of the doublet model (deviance), as well as the optimal mixing ratio. To measure the evidence in favor of a cell being a doublet, we use the difference of deviance ratios of the best-fit doublet and singlet models (equivalent to the log likelihood ratio of the doublet and singlet models, normalized by the log likelihood ratio between the saturated and null models).

**Classifying low-quality cells and doublets**. To identify low-quality cells and classify doublets, we first remove cells that have a high or low proportion of UMIs from mitochondrial genes (>0.25 or <0.01) or with reads at <50 of the reference SNP sites. In many experiments, we observed groups of cells with distinct gene expression profiles, and SNP profiles that did not match to any reference cell line (or pairwise combination of cell lines) in particular, but rather resembled more a mixture of SNPs from all the in-pool cell lines, suggesting that these were empty droplets containing ambient mRNA in the pool[19,23]. To identify these putative empty droplets, we first clustered the single-cell expression profiles using Seurat's default graph-based clustering with 10 nearest neighbors and a resolution parameter of 1–4 (depending on the pool size). We then identified gene expression clusters that consistently had poor-fitting SNP models (i.e., that did not resemble singlets or doublets based on their SNPs). For this, the overall SNP model goodness of fit for each cell was assessed by the deviance ratio of the doublet model, which was strictly greater than or equal to that of the restricted singlet model. The median SNP model deviance ratio was computed for each gene expression cluster, and clusters with a median deviance ratio of <0.3 were considered to be low quality and were removed from the data before further analysis.

We then separated doublets from singlets using a two-component Gaussian mixture model (GMM) fit with two features: the singlet model deviance ratio, and the doublet model goodness of fit (difference in deviance ratios relative to the singlet model). GMMs were fit using the R package MClust[43], with the default conjugate prior on the covariance matrices, and no shrinkage on the component means. Cells with a probability >0.5 of being doublets were then taken to be doublets.

Finally, to ensure cells labeled singlets were confidently identified, we also required that the difference in goodness of fit between the best-fitting and second-best-fitting reference cell lines was at least 2 z-score. Cells that were excluded based on any of the above criteria (other than doublets) were labeled "low quality" (Supplementary Fig. 1).

**Visualizing single-cell expression profiles**. 2D representations of single-cell expression profiles (e.g., Fig. 1b) were generated using Seurat v3[44]. Single-cell counts data were first normalized and log-transformed using the NormalizeData function, with a scale_factor of $10^5$. Data were then normalized across cells using the ScaleData function. The top 5000 most variable genes (based on the "vst" selection method) were selected using the *FindVariableGenes* function, and PCs were computed using the *RunPCA* function, retaining the top $2N$ PCs, where $N$ is the number of cell lines in the pool. t-SNE embeddings were computed based on the PCs, using the *RunTSNE* function with a perplexity parameter of 25. UMAP embeddings were computed using the *RunUMAP* Seurat function with 15 nearest neighbors, and a "min.dist" parameter of 0.6–1.0 (default parameters otherwise).

**Gene-expression-based cell line classification**. For comparison with SNP-based cell line classification, we also classified single cells based on the similarity of their gene expression profiles to bulk RNA-seq measurements from the parental cell lines (using the 19Q3 DepMap gene expression data[45]; available at depmap.org). For this analysis, we combined the control datasets for each cell line pool (untreated or DMSO treated).

Rather than comparing each individual cell's expression profile with the bulk RNA-seq data directly, we first derived de-noised estimates of the single-cell expression profiles by clustering cells and then computing the within-cluster average expression profiles. Specifically, single-cell expression profiles were normalized and scaled, followed by PCA, as described above. We then applied Seurat's default graph-based clustering with 10 nearest neighbors and a resolution parameter of 1 (24 cell line pool) or 20 (99 cell line pool). After identifying clusters, we sum-collapsed read counts across cells within each cluster and then transformed the data to log counts per million (with a pseudo-count parameter of 1). These cluster averages were taken as estimates of each single-cell's expression profile.

To compare these single-cell profiles with bulk RNA-seq profiles we first mean-centered each dataset across samples per gene. We then identified the 5000 genes (present in bulk and single-cell datasets) with highest variance across bulk RNA-seq samples. Each cell was then classified according to the cell line whose bulk RNA-seq profile was most correlated (Pearson correlation) across these 5000 genes.

**Differential expression analysis**. To estimate the average transcriptional response of each cell line to a perturbation, we first sum-collapsed the data—summing read counts across cells for each cell line and treatment condition—to produce a bulk RNA-seq style read counts profile for each sample[46,47]. We then computed normalization factors per sample (cell line and condition) using the "TMMwzp" method from the edgeR R package[48] and transformed the profiles to log counts per million (using a "pseudo-count" of 1) using the edgeR function "cpm" before computing the log fold-change (LFC) difference in relative gene abundances between treatment and control conditions.

Differential expression analyses across cell lines was performed using the "limma-trend" pipeline[49,50], applied to these sum-collapsed and normalized profiles. For this analysis, we included data from cells both 6 and 24 h posttreatment with vehicle control (DMSO) in the control group, as we did not observe a consistent time-related effect of DMSO treatment in our data (e.g., Supplementary Fig. 17). Global differences between the two control conditions were incorporated into the model to help mitigate batch effects[47].

To identify the average drug response across cell lines, we thus used models of the form:

$$Y_{gik} = \beta_g I_k + c_{gk} + b_{jg}, \tag{5}$$

where $Y_{gjk}$, the logCPM expression level of gene $g$ in cell line $j$ and condition $k$, is modeled as a sum of several terms. The first term captures the average treatment effect, where $\beta_g$ is the average LFC of gene $g$ in response to treatment and $I_k$ is an indicator variable representing whether condition $k$ is treatment or control. The second term captures differences in average expression across the control conditions, and the final term captures the baseline expression of each cell line.

To estimate the viability-related and viability-independent response components, we used a similar modeling approach, including the measured drug sensitivity of each cell line as a covariate interacting with treatment as follows:

$$Y_{gjk} = \beta_{0g}I_k + s_j I_k \beta_{1g} + c_{gk} + b_{jg}, \qquad (6)$$

where $S_j$ is the measured sensitivity of cell line $j$ to the treatment ($1 -$ the area under the dose–response curve (AUC)), $\beta_{0g}$ is the viability-independent response of gene $g$ to treatment, and $\beta_{1g}$ is the viability-related response of gene $g$ to treatment.

Only genes with at least 5 reads detected (summed across cells) in at least 5% of the samples were included in analysis. $p$ values were derived from empirical Bayes moderated $t$-statistics, and FDR-adjusted $p$ values were obtained using the Benjamini–Hochberg method[51].

When comparing the transcriptional responses of two cell lines or cell populations to a drug (e.g., Supplementary Figs. 15 and 19), the above method cannot be applied as there would be a single sample for each condition. Hence, we compared the uncollapsed single-cell expression profiles. Specifically, we used the edgeR quasi-likelihood approach[52], following the pipeline used in ref. [53], including cell detection rate (the fraction of genes with non-zero reads detected) as a covariate.

**Drug sensitivity data**. Cell line drug sensitivity data were taken from the Sanger GDSC dataset[4,7], as well as data generated using the PRISM multiplexed drug screening platform[17,20]. For most compounds, we used AUC to measure sensitivity. When data were available from both PRISM and GDSC datasets for a given drug, we used the average of each cell line's AUC values, after quantile normalization of the AUC measurements from each dataset.

For nutlin treatment, we combined nutlin-3a data from GDSC with PRISM data for the nutlin family compound idasanutlin (RG7388). For the tool compound BRD-3379, we found that the (PRISM) data were most reliable for the highest dose, so we used log viability measurements at a single dose of 10 μM, though results were similar when using the AUC.

**Gene set enrichment analysis**. For analysis of gene set enrichment of transcriptional response signatures, we used a simple approach, measuring the set overlap (Fisher's exact test) between each gene set and the 50 top upregulated and downregulated genes across (based on the estimated LFC). The collection of gene sets used was the combination of the "Hallmark" and "Canonical" gene set collections from MSigDB v6.2[54].

**Estimating relative cell line abundance**. Estimates of the effects of perturbations on relative cell line abundance were obtained by counting the number of (QC-passing) single cells from each cell line in each treatment condition, adding a "pseudo-count" of 1, and normalizing counts across cell lines per condition. These relative abundance estimates were averaged across samples for each treatment condition to compute the $\log_2$-fold-change difference between drug-treated and control relative cell line abundances.

**Cell cycle analysis**. Cell cycle phase classification was performed with the Seurat function CellCycleScoring, using the S- and G2M-phase gene lists reported in ref. [29]. The change in proportion of cells in each phase between treatment and control conditions, along with associated confidence intervals, were estimated using the prop.test R function for each cell line. For Fig. 4c, we computed aggregate scores representing how each compound altered the cell cycle composition by computing weighted averages across cell lines of the change in proportion of cells in each phase, where the weights were determined by the cell lines' measured drug sensitivity ($1 -$ AUC, bounded between 0 and 1).

**Principal component analysis**. For PCA (and other machine learning analyses), we used a slightly different procedure to estimate each cell line's average transcriptional response to drug treatment. Rather than sum-collapsing the read count data, we mean-collapsed the single-cell gene expression profiles by normalizing each single-cell profile to counts per million, averaging across cells, and then log-transforming the averaged profiles (using a larger pseudo-count value of 10 to help stabilize LFC estimates for lowly expressed genes). PCA was then computed on the matrix of cell line LFC profiles, mean-centered per gene, using the 5000 genes with most across-cell-line variance. We only used cell lines where there were at least 10 cells in both control and treatment conditions.

The use of mean-collapsed, rather than sum-collapsed, profiles for machine learning analysis helped prevent any bias in the estimated LFC responses related to the number of cells recovered for each cell line. Both sum-collapsed and mean-collapsed LFC estimates produced similar results, differing primarily in whether cells with greater sequencing depth are given more weight.

Comparisons of PC1 loadings with measured drug sensitivity across cell lines (Supplementary Fig. 11) were made using Pearson correlations, with $p$ values estimated using the "cor.test" R function. FDR-adjusted $p$ values were estimated using the Benjamini–Hochberg method[51].

**Transcriptional response embedding**. To compute the embedding of transcriptional response profiles (Fig. 3g), we used the UMAP method[28], as implemented in the Seurat package. Specifically, we compiled all LFC response profiles across cell lines and treatment conditions (computed using mean-collapsed profiles). We restricted analysis to response profiles supported with at least 10 cells per condition and 40 cells in total. We then took the 5000 genes with highest variance across the selected profiles and computed the top 25 PCs. UMAP was then run using cosine distance between samples in this PC space, with an "n.neighbors" parameter of 15 and "min.dist" of 0.7.

**Predictive modeling analysis**. To assess how well we could predict a cell line's drug sensitivity from baseline features or measured transcriptomic responses, we used random forest regression models (implemented in the R package *ranger*[55]) with default parameters. Prediction accuracy ($R^2$ of model predictions) was evaluated using tenfold cross-validation. AUC values were capped at 1.5 before model training to mitigate the effects of a few outliers with large AUC values, though results were similar without capping of AUC values. To help mitigate overfitting, we also applied a pre-filtering of the features, selecting the top 1000 features based on the magnitude of their marginal correlation with the response variable (feature selection was performed separately for each cross-validation set, using training data only). We only included cell lines with at least five cells per condition (treatment and control) for a given drug to ensure that the estimated transcriptional response profiles were sufficiently robust.

To estimate the importance of transcriptional response features used by the model (Fig. 3b), we utilized the "impurity" feature importance metric of the *ranger* package and did not apply pre-selection of features.

For the baseline omics features, we used baseline logTPM expression levels of each protein coding gene, as well as the damaging and hotspot missense mutation status of each gene[6,27]. These data were taken from the DepMap 19Q3 data release[45], available at depmap.org.

**Time course analysis**. Classification of single-cell treatment conditions, as well as doublet classification, from the hashtag read counts data was performed using DemuxEM[56], with default parameters.

We used the same approach described above to estimate the viability-related and viability-independent components of the response at each time point post-trametinib treatment. Since we did not observe substantial transcriptional changes across time points after DMSO treatment (Supplementary Fig. 17), we pooled together data across DMSO conditions for analysis.

For Fig. 5e, f, we plotted the time course of viability-independent and viability-related responses for the top ten downregulated genes in each component, taking the coefficient with the largest magnitude across posttreatment time points for each gene (after filtering for coefficients with FDR < 0.1).

**Bortezomib heterogeneity analysis**. To identify subpopulations of bortezomib-treated cells (Fig. 4d–f), we used Seurat's default methods to normalize the data, detect variable genes, and compute PCs (using 5000 most variable genes and 50 PCs). We then used Seurat's default clustering methods to cluster cells for each cell line (using 20 nearest neighbors and a clustering resolution parameter of 0.25).

**Trametinib heterogeneity analysis**. In order to identify subpopulations of cells from a given cell line in a consistent fashion across baseline and treatment conditions (for analysis in Supplementary Fig. 15), we used the following procedure. After restricting to cells from the target cell line, we used the scTransform method[57] to normalize the data, identify variable genes (we used 5000 genes), and regress out experimental condition as a covariate in order to align clusters across conditions. We then used Seurat's default clustering methods (using the top 10 PCs, 20 nearest neighbors, and a clustering resolution parameter of 0.25) to identify clusters jointly across treated and control cells.

**GPX4 analysis**. Differential expression analysis of GPX4 KO was done by comparing the average effects of the two GPX4 targeting guides against the average of the two control guides (one targeting and one non-targeting), following the same analysis procedure as used for drug treatment data. We identified GPX4-dependent and non-dependent lines using the estimated probability of GPX4 dependency for each cell line from the Achilles 19Q3 "gene dependency" file[45]. Cell lines with GPX4-dependency probability >0.5 were considered dependent.

**L1000 comparison**. L1000 gene expression signatures were taken from either the LINCS Phase 2 data (GEO accession GSE70138, downloaded from http://amp.pharm.mssm.edu/Slicr) or LINCS Phase 1 data (GEO accession GSE92742, downloaded from clue.io). Phase 2 data were used when available (for the compounds trametinib, everolimus, and JQ1), while Phase 1 data were used for the

remaining compounds (bortezomib, gemcitabine, and navitoclax). Comparisons were made using the average of the L1000 Level 5 gene expression signatures across all samples for a given drug and the average LFC values across all cell lines from the MIX-Seq experiment for that drug (24 h posttreatment).

**Reporting summary**. Further information on research design is available in the Nature Research Reporting Summary linked to this article.

## Data availability

All data reported in this manuscript, including single-cell RNA-sequencing data, drug sensitivity measures, and other cell line features used in the analysis, are available on figshare at https://figshare.com/s/139f64b495dea9d88c70[58]. Additional data used in the analysis are also publicly available. Baseline cell line omics features and CRISPR genetic-dependency data are taken from the 19Q3 DepMap dataset[45], available at depmap.org or from figshare at https://figshare.com/articles/DepMap_19Q3_Public/9201770/2. The cell line drug sensitivity data was taken from the Sanger GDSC dataset[4,7], which is available for download from depmap.org or https://www.cancerrxgene.org/, and data generated using the PRISM multiplexed drug screening platform[17,20], which is available for download from depmap.org. The L1000 gene expression signatures were taken from either the LINCS Phase 2 data (GEO accession GSE70138, downloaded from http://amp.pharm.mssm.edu/Slicr) or LINCS Phase 1 data (GEO accession GSE92742, downloaded from clue.io).

## Code availability

Custom code used in the analysis, and for generating all figures, is available at https://github.com/broadinstitute/mix_seq_ms. Code used for SNP classification is available at https://github.com/broadinstitute/single_cell_classification.

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

## Acknowledgements

This work was funded in part by the Broad SPARC program (to F.V., A.T.), Lustgarten Foundation (to A.J.A., B.M.W.), Dana-Farber Cancer Institute Hale Center for Pancreatic Cancer Research (to A.J.A., B.M.W.), the Doris Duke Charitable Foundation (to A.J.A.), Pancreatic Cancer Action Network (to A.J.A.), NIH-NCI K08 CA218420-02 (to A.J.A.), P50CA127003 (to A.J.A.), and U01 CA224146 (to A.J.A.).

## Author contributions

Conceptualization and design: J.M.M., B.R.P., A.W., K.G.-S., T.S., M.R., I.T., A.R., A.J.A., F.V., A.T.; experiments: B.R.P., K.G.-S., T.S., M.R., O.K., D.D., S.B.; data analysis: J.M.M., A.W., W.C., A.J., M.G.; writing: J.M.M., B.R.P., A.W., A.R., A.J.A., F.V., A.T.; supervision: O.R.-R., J.A.R., T.R.G., A.R., A.J.A., F.V., A.T.; project administration: E.C., D.D.; funding acquisition: F.V., A.T., A.J.A., B.M.W., T.R.G., A.R.

## Competing interests

A.R. is a co-founder and equity holder of Celsius Therapeutics, equity holder in Immunitas, and an SAB Member of ThermoFisher Scientific, Neogene Therapeutics, Asimov, and Syros Pharmaceuticals. A.J.A. has consulted for Oncorus, Inc., Arrakis Therapeutics, and Merck & Co., Inc. and has research funding from Mirati Therapeutics and Deerfield Management that are unrelated to this work. A.T. is a consultant for Tango Therapeutics. T.R.G. is a consultant to GlaxoSmithKline, a founder of Sherlock Biosciences, and was formerly a consultant and equity holder in Foundation Medicine, acquired by Roche. T.R.G. also receives research funding unrelated to this project from Bayer Healthcare. B.M.W. receives research funding unrelated to this project from Celgene and Lilly and is a consultant for BioLineRx, Celgene, G1 Therapeutics, and GRAIL. The Broad Institute filed a United States Provisional Patent Application directed to work described in this article. The pending Application is entitled "Rapid Prediction of Drug Responsiveness" and was filed on September 2, 2019. The current inventor list includes A.T., A.J.A., F.V., B.R.P., and J.M.M. A.W., K.G.-S., T.S., M.R., O.K., W.N.C., A.J., E.C., D.D., S.B., M.G., I.T., O.R.-R., and J.A.R. declare no competing interests.
