## [Peer Review File · Nature Communications]

Reviewers' Comments:

Reviewer #1:

Remarks to the Author:

The is a very mature and clearly written manuscript. The authors present an approach for SNP based demultiplexing of single cell RNA-seq data to measure transcriptional effects of perturbations. This is a very active research field and the contribution is very timely. The methods proposed and used are all very solid and state-of-the art.

I really like the didactic way of presenting the linear model in Fig2, which really explains what is being modelled here.

I only have two minor comments:

- for consistency, would it hurt to decide on either tSNE or UMAP for visualisation, rather than switching from one to the other between figures?
- in the methods section, in the multiplexing model, the authors use a logistic regression model - this is a supervised model, so on top of the X_{ij} you also need a label for each SNP i (=alternative allele or not). I did not see a clear description on how these labels (= the ground truth) were assigned and ideally would like to see this clarified.

Florian Markowetz

Reviewer #2:

Remarks to the Author:

McFarland et al developed a method (MIX-Seq) to measure post-perturbational transcriptomic response at single-cell resolution multiplexed across a wide range of cell lines by resolving the identity of each cell based on single nucleotide polymorphism (SNP) profiles from mRNA. Utilizing this method, they profiled transcriptomic response of 13 drugs at multiple time points across a range of cancer cell lines characterized previously in DepMap. Based on these profiles, the authors first identified drug-specific and shared transcriptional response signatures that: 1. Are informative about the drug's mechanism of action (MOA), via benchmarking on the subset with known MOAs, 2. Are associated or are independent of sensitivity, 3. could predict long-term viability response, and finally, 4. Could predict heterogeneity of the response. Finally, combining this approach with cell-hashing they performed dual-multiplexing across time points or dosages.

Overall, this is a well-written manuscript describing the experimental and statistical methods clearly and briefly.

Comments:

1. An additional validation for resolving the identity of each cell based on SNP profiles from mRNA could be performed by integrating pre/post- treatment transcriptomic profiles from CMAP (Subramanian et al 2017, for ~20K small molecules for 1000 genes across 9 cancer cell lines from DepMap across two time points and multiple dosages) for the subset drugs and cell lines used in this study that are common in CMAP. In some more detail, we propose: (for every drug separately) Each cell transcriptomic response from MIX-seq can be compared based on the similarity (by a correlation test) to CMAP-based nine cell lines transcriptomic response for the matched drug. Identification of cell lines (out of nine) can be done based on this correlation rho which can act as labels to compare against SNP-based method predictions. Importantly this framework could be used to integrate other CMAP like datasets like DRUG-seq (Chaoyang Ye et al 2018) and PLATE-seq (Bush, E. C. et 2017).

2. It would also help to compare the long-term viability prediction power of treatment response

transcriptome profile from single-cell seq based MIX-seq vs bulk-seq based from CMAP.

3. The last section highlights a later-onset regulation of viability-related genes vs viability-independent genes. We would like to point out that viability-related genes could have a later onset due to the identification methodology rather of underlying biology. Viability-related genes are differentially expressed in sensitive vs non-sensitive cell lines. This is because when the sensitive cells subset is targeted it leads to a population decrease with treatment time at a higher extent than non-sensitive cell lines. Thus, the extent of differential expression of viability-related would increase with time. Hence, need to be corrected for during this analysis.

Reviewer #1 (Remarks to the Author):

The is a very mature and clearly written manuscript. The authors present an approach for SNP based demultiplexing of single cell RNA-seq data to measure transcriptional effects of perturbations. This is a very active research field and the contribution is very timely. The methods proposed and used are all very solid and state-of-the art.

I really like the didactic way of presenting the linear model in Fig2, which really explains what is being modelled here.

We thank Dr. Markowitz for these positive comments.

I only have two minor comments:

- for consistency, would it hurt to decide on either tSNE or UMAP for visualisation, rather than switching from one to the other between figures?

Thank you for pointing out this source of inconsistency. In general we have found that tSNE and UMAP produce qualitatively similar results, though with some slight differences in their ability to capture different features of the data. Most notably, UMAP better captures the time-dependent variation within each cluster for the time-course analysis in Fig. 5b. Thus, in the revised manuscript we have adopted UMAP throughout for visualization of single-cell expression profiles. The only figure where we retain tSNE for visualization in the revised manuscript is Supplementary Fig. 16a, where we plot a 2d representation of the hash tag read-count profiles, as we needed to provide a custom distance matrix for this analysis – functionality which is not readily available in the UMAP implementation we use.

- in the methods section, in the multiplexing model, the authors use a logistic regression model - this is a supervised model, so on top of the X_{ij} you also need a label for each SNP i (=alternative allele or not). I did not see a clear description on how these labels (= the ground truth) were assigned and ideally would like to see this clarified.

We thank Dr. Markowitz for this comment and realize the need for further clarity regarding what the X_{ij} represents in our model. The vectors \mathbf{X}_j represent the precomputed SNP profiles of each reference cell line j . Each entry, X_{ij} , in these vectors thus represents the fraction of reads from SNP site i observed from the alternate allele in reference cell line j . These are computed based on bulk RNA-seq data. We have revised the text in the Methods section (page 19, “SNP-based cell line classification”) to help clarify this point.

Florian Markowitz

Reviewer #2 (Remarks to the Author):

McFarland et al developed a method (MIX-Seq) to measure post-perturbational transcriptomic response at single-cell resolution multiplexed across a wide range of cell lines by resolving the identity of each cell based on single nucleotide polymorphism (SNP) profiles from mRNA. Utilizing this method, they profiled transcriptomic response of 13 drugs at multiple time points across a range of cancer cell lines characterized previously in DepMap. Based on these profiles, the authors first identified drug-specific and shared transcriptional response signatures that: 1. Are informative about the drug's mechanism of action (MOA), via benchmarking on the subset with known MOAs, 2. Are associated or are independent of sensitivity, 3. could predict long-term viability response, and finally, 4. Could predict heterogeneity of the response. Finally, combining this approach with cell-hashing they performed dual-multiplexing across time points or dosages.

Overall, this is a well-written manuscript describing the experimental and statistical methods clearly and briefly.

We thank the reviewer for this positive summary.

Comments:

1. An additional validation for resolving the identity of each cell based on SNP profiles from mRNA could be performed by integrating pre/post- treatment transcriptomic profiles from CMAP (Subramanian et al 2017, for ~20K small molecules for 1000 genes across 9 cancer cell lines from DepMap across two time points and multiple dosages) for the subset drugs and cell lines used in this study that are common in CMAP. In some more detail, we propose: (for every drug separately) Each cell transcriptomic response from MIX-seq can be compared based on the similarity (by a correlation test) to CMAP-based nine cell lines transcriptomic response for the matched drug. Identification of cell lines (out of nine) can be done based on this correlation rho which can act as labels to compare against SNP-based method predictions. Importantly this framework could be used to integrate other CMAP like datasets like DRUG-seq (Chaoyang Ye et al 2018) and PLATE-seq (Bush, E. C. et 2017).

We thank the reviewer for highlighting the potential for comparing our MIX-Seq results to other published transcriptional profiling datasets. In general, we agree that such comparisons will help strengthen the manuscript (see additional details below).

For the purpose of validating the SNP-based cell classification, we believe that it is more straightforward to use the 'baseline' transcriptional profiles, rather than the post-treatment transcriptional responses. Indeed, we used independent measurements of the cell lines' baseline expression profiles (using bulk RNA-seq) to classify cells and found nearly perfect agreement with our SNP-based cell classification (Supplementary Fig. 2).

Using post-perturbational responses for validation of the cell classification presents additional technical challenges. For example, the CMAP L1000 dataset (Subramanian et al., 2017) is normalized so that the measurements reflect the perturbation-induced change in a given cell line's expression profile relative to baseline. In our case, we aren't able to estimate differential responses (treatment vs control) for each individual cell in order to make direct

comparisons – rather we can only estimate perturbation responses on a per cell line basis, after assigning individual cells from treatment and control conditions to the appropriate cell line.

Following this general suggestion by the reviewer, we performed systematic comparisons of the drug-induced transcriptional response data generated by MIX-Seq with data available from CMAP (using the L1000 platform). These results (revised Supplementary Fig. 5) show good overall agreement between the two methods, and provide additional validation of transcriptional response profiling with MIX-Seq. We also explored comparisons with additional datasets such as DRUG-seq and PLATE-seq, but found that there was minimal overlap of drugs and cell lines used making direct comparisons difficult.

2. It would also help to compare the long-term viability prediction power of treatment response transcriptome profile from single-cell seq based MIX-seq vs bulk-seq based from CMAP.

Again, we thank the reviewer for highlighting this potential connection with existing datasets. Indeed, recent work by our group and others (Szalai et al, *Nucleic Acids Research*, 2019; Jones et al, *bioRxiv* 2020) have performed related analyses using the CMAP data. An important limitation of this previous work, however, is that the CMAP data generally includes profiles for each compound (or genetic perturbation) across a small handful of cell lines. While there are a large number of perturbations profiles, this limits the power to assess viability-prediction power for a given compound. Hence these studies largely evaluated this question by pooling data across available drugs. When we used an analogous approach of pooling the MIX-Seq data across drugs, we found somewhat higher predictive accuracy ($R^2 = 0.5$; results described in Fig. 3b) compared with that found in these previous studies using the CMAP data (R^2 of 0.1-0.35), though we note that direct comparisons are difficult to interpret due to differences in the compounds used as well as other experimental differences.

In the revised Discussion section we have added additional text (page 15, paragraph 3) highlighting these related efforts using CMAP data, as well as the relative tradeoffs provided by MIX-Seq's ability to profile responses across many cell lines for a given compound.

3. The last section highlights a later-onset regulation of viability-related genes vs viability-independent genes. We would like to point out that viability-related genes could have a later onset due to the identification methodology rather of underlying biology. Viability-related genes are differentially expressed in sensitive vs non-sensitive cell lines. This is because when the sensitive cells subset is targeted it leads to a population decrease with treatment time at a higher extent than non-sensitive cell lines. Thus, the extent of differential expression of viability-related would increase with time. Hence, need to be corrected for during this analysis.

We fully agree with the reviewer that this is an important consideration with these analyses. Indeed, it was for this reason that we chose to perform our viability-prediction analyses using transcriptional response signatures computed by *averaging* normalized expression profiles across cells per cell line, rather than *summing* the read counts across cells as we do in most of our analyses. This ensures that the resultant expression profiles are unbiased by the number of cells profiled in each cell line.

The reviewer makes a very good point that it's important to ensure that the differential time-course of viability-related and viability-independent response components (shown in Fig. 5) is also unbiased by variable cell numbers. We have thus added additional analyses

(Supplementary Fig. 18 in the revised manuscript) showing that we find very similar results when using the unbiased cell-averaging approach to quantify transcriptional responses, rather than summing reads across cells.

Reviewers' Comments:

Reviewer #1:

Remarks to the Author:

I was already very positive in the first round and the paper only has improved since then.

Florian Markowetz

Reviewer #2:

Remarks to the Author:

Revision of McFarland et al 2020

--Authors have performed all the additional analysis suggested to further strengthen their findings in an appropriate and satisfactory manner. We recommend this study for acceptance in the current form.

McFarland et al developed a method (MIX-Seq) to measure post-perturbational transcriptomic response at single-cell resolution multiplexed across a wide range of cell lines by resolving the identity of each cell based on single nucleotide polymorphism (SNP) profiles from mRNA. Utilizing this method, they profiled transcriptomic response of 13 drugs at multiple time points across a range of cancer cell lines characterized previously in DepMap. Based on these profiles, the authors first identified drug-specific and shared transcriptional response signatures that: 1. Are informative about the drug's mechanism of action (MOA), via benchmarking on the subset with known MOAs, 2. Are associated or are independent of sensitivity, 3. could predict long-term viability response, and finally, 4. Could predict heterogeneity of the response. Finally, combining this approach with cell-hashing they performed dual-multiplexing across time points or dosages.

Overall, this is a well-written manuscript describing the experimental and statistical methods clearly and briefly.

We thank the reviewer for this positive summary.

Comments:

1. An additional validation for resolving the identity of each cell based on SNP profiles from mRNA could be performed by integrating pre/post- treatment transcriptomic profiles from CMAP (Subramanian et al 2017, for ~20K small molecules for 1000 genes across 9 cancer cell lines from DepMap across two time points and multiple dosages) for the subset drugs and cell lines used in this study that are common in CMAP. In some more detail, we propose: (for every drug separately) Each cell transcriptomic response from MIX-seq can be compared based on the similarity (by a correlation test) to CMAP-based nine cell lines transcriptomic response for the matched drug. Identification of cell lines (out of nine) can be done based on this correlation rho which can act as labels to compare against SNP-based method predictions. Importantly this framework could be used to integrate other CMAP like datasets like DRUG-seq (Chaoyang Ye et al 2018) and PLATE-seq (Bush, E. C. et 2017).

We thank the reviewer for highlighting the potential for comparing our MIX-Seq results to other published transcriptional profiling datasets. In general, we agree that such comparisons will help strengthen the manuscript (see additional details below).

For the purpose of validating the SNP-based cell classification, we believe that it is more straightforward to use the 'baseline' transcriptional profiles, rather than the post-treatment transcriptional responses. Indeed, we used independent measurements of the cell lines' baseline expression profiles (using bulk RNA-seq) to classify cells and found nearly perfect agreement with our SNP-based cell classification (Supplementary Fig. 2).

Using post-perturbational responses for validation of the cell classification presents additional

technical challenges. For example, the CMAP L1000 dataset (Subramanian et al., 2017) is normalized so that the measurements reflect the perturbation-induced change in a given cell line's expression profile relative to baseline. In our case, we aren't able to estimate differential responses (treatment vs control) for each individual cell in order to make direct comparisons – rather we can only estimate perturbation responses on a per cell line basis, after assigning individual cells from treatment and control conditions to the appropriate cell line.

Following this general suggestion by the reviewer, we performed systematic comparisons of the drug-induced transcriptional response data generated by MIX-Seq with data available from CMAP (using the L1000 platform). These results (revised Supplementary Fig. 5) show good overall agreement between the two methods, and provide additional validation of transcriptional response profiling with MIX-Seq. We also explored comparisons with additional datasets such as DRUG-seq and PLATE-seq, but found that there was minimal overlap of drugs and cell lines used making direct comparisons difficult.

--We agree with the author that using baseline expression to identify cell identity is the most optimum way here. It is also encouraging to see an overall agreement of MIX-seq and CMAP.

2. It would also help to compare the long-term viability prediction power of treatment response transcriptome profile from single-cell seq based MIX-seq vs bulk-seq based from CMAP.

Again, we thank the reviewer for highlighting this potential connection with existing datasets. Indeed, recent work by our group and others (Szalai et al, *Nucleic Acids Research*, 2019; Jones et al, *bioRxiv* 2020) have performed related analyses using the CMAP data. An important limitation of this previous work, however, is that the CMAP data generally includes profiles for each compound (or genetic perturbation) across a small handful of cell lines. While there are a large number of perturbations profiles, this limits the power to assess viability-prediction power for a given compound. Hence these studies largely evaluated this question by pooling data across available drugs. When we used an analogous approach of pooling the MIX-Seq data across drugs, we found somewhat higher predictive accuracy ($R^2 = 0.5$; results described in Fig. 3b) compared with that found in these previous studies using the CMAP data (R^2 of 0.1-0.35), though we note that direct comparisons are difficult to interpret due to differences in the compounds used as well as other experimental differences.

In the revised Discussion section we have added additional text (page 15, paragraph 3) highlighting these related efforts using CMAP data, as well as the relative tradeoffs provided by MIX-Seq's ability to profile responses across many cell lines for a given compound.

--The analysis performed is satisfactory.

3. The last section highlights a later-onset regulation of viability-related genes vs viability-independent genes. We would like to point out that viability-related genes could have a later onset due to the identification methodology rather of underlying biology. Viability-related genes are differentially expressed in sensitive vs non-sensitive cell lines. This is because when the sensitive cells subset is targeted it leads to a population decrease with treatment time at a higher extent than non-sensitive cell lines. Thus, the extent of differential expression of viability-related would increase with time. Hence, need to be corrected for during this analysis.

We fully agree with the reviewer that this is an important consideration with these analyses. Indeed, it was for this reason that we chose to perform our viability-prediction analyses using transcriptional response signatures computed by averaging normalized expression profiles across cells per cell line, rather than summing the read counts across cells as we do in most of our analyses. This ensures that the resultant expression profiles are unbiased by the number of cells profiled in each cell line.

The reviewer makes a very good point that it's important to ensure that the differential time-course of viability-related and viability-independent response components (shown in Fig. 5) is also unbiased by variable cell numbers. We have thus added additional analyses (Supplementary Fig. 18 in the revised manuscript) showing that we find very similar results when using the unbiased cell-averaging approach to quantify transcriptional responses, rather than summing reads across cells.

--Again, the analysis performed is satisfactory.

Overall the reviewers were satisfied with our responses to their initial questions and concerns, and did not describe any remaining issues.

Comments:

1. An additional validation for resolving the identity of each cell based on SNP profiles from mRNA could be performed by integrating pre/post- treatment transcriptomic profiles from CMAP (Subramanian et al 2017, for ~20K small molecules for 1000 genes across 9 cancer cell lines from DepMap across two time points and multiple dosages) for the subset drugs and cell lines used in this study that are common in CMAP. In some more detail, we propose: (for every drug separately) Each cell transcriptomic response from MIX-seq can be compared based on the similarity (by a correlation test) to CMAP-based nine cell lines transcriptomic response for the matched drug. Identification of cell lines (out of nine) can be done based on this correlation rho which can act as labels to compare against SNP-based method predictions. Importantly this framework could be used to integrate other CMAP like datasets like DRUG-seq (Chaoyang Ye et al 2018) and PLATE-seq (Bush, E. C. et 2017).

We thank the reviewer for highlighting the potential for comparing our MIX-Seq results to other published transcriptional profiling datasets. In general, we agree that such comparisons will help strengthen the manuscript (see additional details below).

For the purpose of validating the SNP-based cell classification, we believe that it is more straightforward to use the 'baseline' transcriptional profiles, rather than the post-treatment transcriptional responses. Indeed, we used independent measurements of the cell lines' baseline expression profiles (using bulk RNA-seq) to classify cells and found nearly perfect agreement with our SNP-based cell classification (Supplementary Fig. 2).

Using post-perturbational responses for validation of the cell classification presents additional technical challenges. For example, the CMAP L1000 dataset (Subramanian et al., 2017) is normalized so that the measurements reflect the perturbation-induced change in a given cell line's expression profile relative to baseline. In our case, we aren't able to estimate differential responses (treatment vs control) for each individual cell in order to make direct comparisons – rather we can only estimate perturbation responses on a per cell line basis, after assigning individual cells from treatment and control conditions to the appropriate cell line.

Following this general suggestion by the reviewer, we performed systematic comparisons of the drug-induced transcriptional response data generated by MIX-Seq with data available from CMAP (using the L1000 platform). These results (revised Supplementary Fig. 5) show good overall agreement between the two methods, and provide additional validation of transcriptional response profiling with MIX-Seq. We also explored comparisons with additional datasets such as DRUG-seq and PLATE-seq, but found that there was minimal overlap of drugs and cell lines used making direct comparisons difficult.

--We agree with the reviewer that using baseline expression to identify cell identity is the most optimum way here. It is also encouraging to see an overall agreement of MIX-seq and CMAP.

We thank the reviewer for this positive comment.

2. It would also help to compare the long-term viability prediction power of treatment response transcriptome profile from single-cell seq based MIX-seq vs bulk-seq based from CMAP. Again, we thank the reviewer for highlighting this potential connection with existing datasets. Indeed, recent work by our group and others (Szalai et al, Nucleic Acids Research, 2019; Jones et al, bioRxiv 2020) have performed related analyses using the CMAP data. An important limitation of this previous work, however, is that the CMAP data generally includes profiles for each compound (or genetic perturbation) across a small handful of cell lines. While there are a large number of perturbations profiles, this limits the power to assess viability-prediction power for a given compound. Hence these studies largely evaluated this question by pooling data across available drugs. When we used an analogous approach of pooling the MIX-Seq data across drugs, we found somewhat higher predictive accuracy ($R^2 = 0.5$; results described in Fig. 3b) compared with that found in these previous studies using the CMAP data (R^2 of 0.1-0.35), though we note that direct comparisons are difficult to interpret due to differences in the compounds used as well as other experimental differences. In the revised Discussion section we have added additional text (page 15, paragraph 3) highlighting these related efforts using CMAP data, as well as the relative tradeoffs provided by MIX-Seq's ability to profile responses across many cell lines for a given compound.
--The analysis performed is satisfactory.

We thank the reviewer for this positive response.

3. The last section highlights a later-onset regulation of viability-related genes vs viability-independent genes. We would like to point out that viability-related genes could have a later onset due to the identification methodology rather of underlying biology. Viability-related genes are differentially expressed in sensitive vs non-sensitive cell lines. This is because when the sensitive cells subset is targeted it leads to a population decrease with treatment time at a higher extent than non-sensitive cell lines. Thus, the extent of differential expression of viability-related would increase with time. Hence, need to be corrected for during this analysis. We fully agree with the reviewer that this is an important consideration with these analyses. Indeed, it was for this reason that we chose to perform our viability-prediction analyses using transcriptional response signatures computed by averaging normalized expression profiles across cells per cell line, rather than summing the read counts across cells as we do in most of our analyses. This ensures that the resultant expression profiles are unbiased by the number of cells profiled in each cell line. The reviewer makes a very good point that it's important to ensure that the differential time-course of viability-related and viability-independent response components (shown in Fig. 5) is also unbiased by variable cell numbers. We have thus added additional analyses (Supplementary Fig. 18 in the revised manuscript) showing that we find very similar results when using the unbiased cell-averaging approach to quantify transcriptional responses, rather than summing reads across cells.
--Again, the analysis performed is satisfactory.

We thank the reviewer for their positive comment.